# Macrophages restrict the nephrogenic field and promote endothelial connections during kidney development

David AD Munro[1]*, Yishay Wineberg[2,3], Julia Tarnick[1], Chris S Vink[4], Zhuan Li[4], Clare Pridans[4], Elaine Dzierzak[4], Tomer Kalisky[2,3], Peter Hohenstein[5,6], Jamie A Davies[1]

[1]Centre for Discovery Brain Sciences, The University of Edinburgh, Edinburgh, United Kingdom; [2]Department of Bioengineering, Bar-Ilan University, Ramat Gan, Israel; [3]Institute of Nanotechnology and Advanced Materials, Bar-Ilan University, Ramat Gan, Israel; [4]Centre for Inflammation Research, Queen's Medical Research Institute, The University of Edinburgh, Edinburgh, United Kingdom; [5]Leiden University Medical Center, Leiden University, Leiden, The Netherlands; [6]The Roslin Institute, The University of Edinburgh, Midlothian, United Kingdom

**Abstract** The origins and functions of kidney macrophages in the adult have been explored, but their roles during development remain largely unknown. Here we characterise macrophage arrival, localisation, heterogeneity, and functions during kidney organogenesis. Using genetic approaches to ablate macrophages, we identify a role for macrophages in nephron progenitor cell clearance as mouse kidney development begins. Throughout renal organogenesis, most kidney macrophages are perivascular and express F4/80 and CD206. These macrophages are enriched for mRNAs linked to developmental processes, such as blood vessel morphogenesis. Using antibody-mediated macrophage-depletion, we show macrophages support vascular anastomoses in cultured kidney explants. We also characterise a subpopulation of galectin-3+ (Gal3+) myeloid cells within the developing kidney. Our findings may stimulate research into macrophage-based therapies for renal developmental abnormalities and have implications for the generation of bioengineered kidney tissues.
DOI: https://doi.org/10.7554/eLife.43271.001

*For correspondence:
david.munro@ed.ac.uk

**Competing interests:** The authors declare that no competing interests exist.

## Introduction

Macrophages are professional phagocytes with roles in tissue development, immunity, regeneration, remodelling, and repair (*Bosurgi et al., 2017*; *Eom and Parichy, 2017*; *Godwin et al., 2013*; *Stamatiades et al., 2016*). In the adult kidney, macrophages can cause, prevent, and repair damage (*Cao et al., 2015*; *Rogers et al., 2014*). During organogenesis, evidence suggests that macrophages promote kidney growth (*Alikhan et al., 2011*; *Rae et al., 2007*), but details regarding their developmental localisation and functions are lacking. Characterising the roles for kidney macrophages may lead to the development of macrophage-based strategies to enhance renal maturation in cases of developmental abnormalities and premature birth.

Kidney development begins when the caudal Wolffian duct is induced to branch, forming a ureteric bud that invades the metanephric mesenchyme by embryonic day (E) 11 in the mouse (*Pichel et al., 1996*; *Saxén and Sariola, 1987*). The ureteric bud undergoes rounds of iterative branching to form the adult collecting duct tree, while cells endogenous to the metanephric mesenchyme generate nephrons and stroma (*Grobstein, 1955*; *Kobayashi et al., 2008*; *Epelman et al., 2014*; *Shakya et al., 2005*; *Short et al., 2014*). Other kidney cell types, such as monocytes,

**eLife digest** The kidneys clean our blood by filtering out waste products while ensuring that useful components, like nutrients, remain in the bloodstream. Blood enters the kidneys through a network of intricately arranged blood vessels, which associate closely with the 'cleaning tubes' that carry out filtration. Human kidneys start developing during the early phases of embryonic development. During this process, the newly forming blood vessels and cleaning tubes must grow in the right places for the adult kidney to work properly.

Macrophages are cells of the immune system that clear away foreign, diseased, or damaged cells. They are also thought to encourage growth of the developing kidney, but how exactly they do this has remained unknown. Munro et al. therefore wanted to find out when macrophages first appeared in the embryonic kidney and how they might help control their development.

Experiments using mice revealed that the first macrophages arrived in the kidney early during its development, alongside newly forming blood vessels. Further investigation using genetically modified mice that did not have macrophages revealed that these immune cells were needed at this stage to clear away misplaced kidney cells and help 'set the scene' for future development.

At later stages, macrophages in the kidney interacted closely with growing blood vessels. As well as producing molecules linked with blood vessel formation, the macrophages wrapped around the vessels themselves, sometimes even eating cells lining the vessels and the blood cells carried within them. These observations suggested that macrophages actively shaped the network of blood vessels developing within the kidneys. Experiments removing macrophages from kidney tissue confirmed this: in normal kidneys, the blood vessels grew into a continuous network, but in kidneys lacking macrophages, far fewer connections formed between the vessels.

This work sheds new light on how the complex structures in the adult kidney first arise and could be useful in future research. For example, adding macrophages to simplified, laboratory-grown 'mini-kidneys' could make them better models to study kidney growth, while patients suffering from kidney diseases might benefit from new drugs targeting macrophages.

DOI: https://doi.org/10.7554/eLife.43271.002

macrophages, and most endothelial cells, derive from extra-renal sources (*Epelman et al., 2014*; *Hoeffel et al., 2015*; *DeFalco et al., 2014*; *Sequeira-Lopez et al., 2015*; *Sims-Lucas et al., 2013*).

In embryonic development, macrophage progenitors arrive in organs in waves. The first wave is generated from yolk sac-derived primitive macrophage progenitors (*Palis et al., 1999*; *Schulz et al., 2012*), the second from yolk sac-derived erythro-myeloid progenitors (EMPs) that migrate and colonise the fetal liver (*Hoeffel et al., 2015*; *Rantakari et al., 2016*), and the third from haematopoietic stem cells (HSCs) that emerge in the aorta-gonad-mesonephros region (*Medvinsky and Dzierzak, 1996*; *Sheng et al., 2015*). Unlike yolk sac-derived primitive macrophage progenitors, both HSCs and EMPs in the foetal liver are thought to pass through a monocytic intermediate phase before differentiating into mature macrophages (*Hoeffel et al., 2015*; *Hoeffel and Ginhoux, 2018*; *Schulz et al., 2012*). Lineage tracing studies in mice suggest that kidney macrophages are initially derived from the yolk sac but are derived almost exclusively from foetal monocytes after birth (*Epelman et al., 2014*; *Hoeffel et al., 2015*; *Munro and Hughes, 2017*; *Sheng et al., 2015*).

In the adult, macrophages and endothelial cells form anatomical and functional units that function to clear immune complexes from the renal blood (*Stamatiades et al., 2016*). In development, kidney macrophages closely associate with the epithelial tubules of nephrons (*Rae et al., 2007*), but their interactions with vascular endothelial cells have not been examined. In recent years, kidney vascularisation has been a focus of renal developmental research (*Daniel et al., 2018*; *Halt et al., 2016*; *Hu et al., 2016*; *Munro et al., 2017a*; *Munro et al., 2017b*); however, possible roles for macrophages in this process remain unexplored. We show that macrophages in the embryonic kidney restrict the early domain of nephron progenitor cells, frequently interact with blood vessels, are enriched for mRNAs linked to vascular development, and promote endothelial cross-connections.

## Results

### Macrophages clear rostral nephron progenitors as kidney development initiates

To characterise macrophage distribution as the metanephric kidney's component parts emerge, we immunostained and optically cleared whole-mount E9.5–10.5 mouse embryos. At these stages, all macrophages in the embryo-proper are derived from the yolk sac and express colony-stimulating factor one receptor (Csf1r), fractalkine receptor (Cx3cr1), and EGF-like module receptor 1 (F4/80) (*Mass et al., 2016*; *Schulz et al., 2012*). Csf1r and Cx3cr1 mark both mature macrophages and their precursors, whereas F4/80 marks only mature macrophages (*Frame et al., 2016*; *Mass et al., 2016*).

At E9.5, bilateral nephrogenic cell populations, which expressed sine oculis-related homeobox 2 (Six2), were arranged as cords that lay medial to Wolffian ducts and lateral to the dorsal aorta in the caudal part of the mouse embryo (*Figure 1a–c*; *Video 1*). No F4/80$^+$ macrophages were near to the Six2$^+$ nephrogenic cords at this stage (*Figure 1c*; *Video 2*). This finding is consistent with previous studies using *Cx3cr1$^{GFP}$* embryos that showed GFP-expressing cells arrived in the embryo-proper only from E9.5 onwards (*Mass et al., 2016*; *Gomez Perdiguero et al., 2013*; *Schulz et al., 2012*; *Stremmel et al., 2018*).

By E10.5, as ureteric bud outgrowth started, Six2$^+$ metanephric cells had accumulated at the caudal end of the Wolffian duct (*Figure 1d*; *Figure 1—figure supplement 1*) (as previously described by *Wainwright et al., 2015*). Immunostaining for a pan-endothelial marker (CD31) demonstrated that the metanephric cell populations were avascular at this stage (*Figure 1e*; *Video 2*). CD31 also marked primitive germ cells and intra-aortic hematopoietic clusters containing HSCs in the caudal part of the E10.5 mouse embryo (*Figure 1—figure supplement 1*; *Gomperts et al., 1994*; *Wakayama et al., 2003*; *Yokomizo and Dzierzak, 2010*). F4/80$^+$ macrophages were now present in high numbers in the caudal part of the embryo (*Figure 1e–f*). These macrophages largely avoided the metanephric mesenchyme (*Figure 1e–f*; *Video 2*) but localised in high numbers alongside rostral clusters of Six2$^+$ cells (*Figure 1g–h*).

The localisation of macrophages at E10.5 suggested to us that they may clear rostral Six2$^+$ cells as the metanephric mesenchyme assembles at the caudal aspect of the Wolffian duct. To test this hypothesis, we depleted macrophages in vivo by crossing transgenic mice, expressing codon-optimised Cre (iCre) under the control of the Csf1r promoter (*Csf1r$^{icre}$*), with transgenic mice that had a floxed-STOP cassette and the diphtheria toxin A subunit (DTA) knocked into the Rosa26 locus (*Rosa$^{DTA}$*) (*Figure 1—figure supplement 2*). In this system, DTA expression is specifically induced in iCre$^+$ cells, resulting in cell death by inhibiting protein synthesis (*Breitman et al., 1987*; *Collier, 2001*; *Plummer et al., 2017*) (*Figure 1—figure supplement 2*). Consistent with the hypothesis that macrophages clear rostral nephrogenic progenitors, macrophage-depleted E11.5 *Csf1ricre$^{icre+}$-Rosa$^{DTA}$* embryos had elongated metanephric mesenchyme populations and larger rostral clusters of Six2$^+$ cells compared to somite pair-matched littermate controls (*Figure 1i–l*). Moreover, Six2$^+$ nuclei of some rostral nephrogenic progenitors were observed within the cell bodies of macrophages, suggesting that active phagocytosis occurs at this site (*Figure 1j*; *Figure 1—figure supplement 3*).

As Six2$^+$ nephrogenic progenitors secrete signals that stimulate ureteric bud outgrowth (*Sainio et al., 1997*), we next examined whether persistence of rostral nephrogenic progenitors in the absence of macrophages resulted in ectopic rostral ureteric bud outgrowth. In all macrophage-depleted embryos, ureteric bud outgrowth had occurred only at the normal anatomical position (in 9/9 kidneys analysed); however, its morphological development was delayed compared to littermate controls in E11.5 and E12.5 embryos (*Figure 1—figure supplement 4*). Vascular organisation appeared normal in macrophage-depleted embryos relative to the developmental stage of the kidney (*Figure 1i*; *Figure 1—figure supplement 4*).

Collectively, these data show that macrophages arrive near to the metanephric mesenchyme as it condenses at the caudal aspect of the Wolffian duct at E9.5-E10.5. These early macrophages arrange alongside rostral nephrogenic cells but avoid caudal ones. In the absence of macrophages, rostral nephrogenic cell clearance and ureteric bud development are delayed.

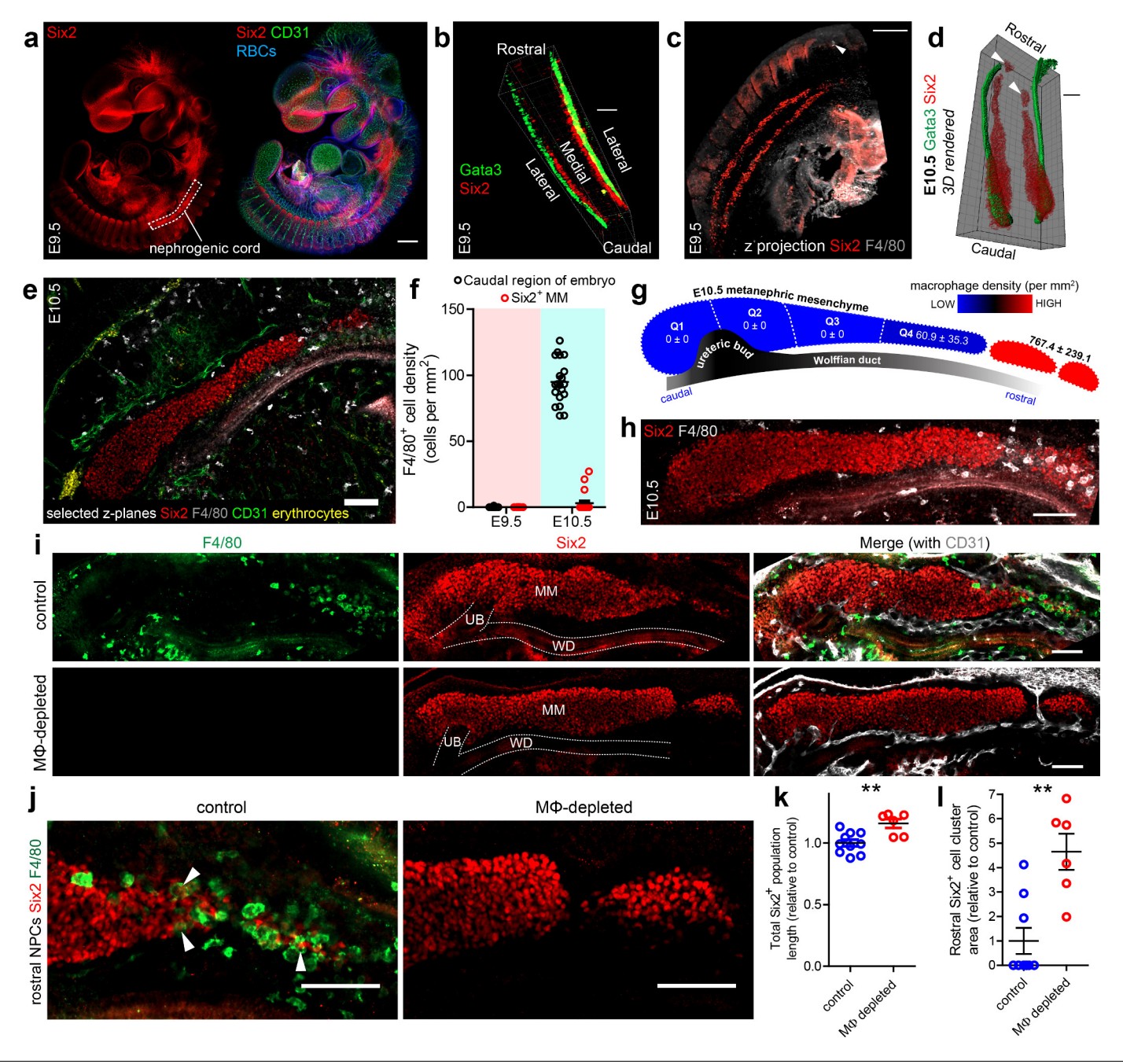

**Figure 1.** Macrophages and the initiation of kidney development. (a) Nephrogenic cord location in the E9.5 embryo. (b) E9.5 nephrogenic cord and Wolffian duct arrangement. (c) Z-projection of E9.5 caudal part of mouse embryo and nephrogenic cords. White arrowhead shows an F4/80[+] macrophage. (d) 3D rendering of E10.5 Wolffian ducts (Gata3) and nephrogenic progenitors (Six2). White arrowheads show isolated clusters of rostral nephrogenic cells. (e) F4/80[+] macrophage localisation relative to metanephric mesenchyme at E10.5. (f) Macrophage density at E9.5 and E10.5 (n = 20 fields of view from two cleared E9.5 embryos and 20 fields from two cleared E10.5 embryos). MM, metanephric mesenchyme. (g–h) Macrophage density along the rostro-caudal axis of the E10.5 metanephric mesenchyme and within isolated groups of rostral nephrogenic cells (n = 4 kidneys from two embryos). Image in (h) shows a representative image of an E10.5 kidney. (i) Representative control and macrophage (MΦ)-depleted E11.5 metanephric mesenchyme populations (n for controls = 10 kidneys from six embryos; n for MΦ-depleted = 6 kidneys from three embryos). (j) Rostral nephrogenic progenitor cells (NPCs) in control and MΦ-depleted embryos. White arrowheads in left panel show Six2[+] nuclei within cell bodies of F4/80[+] macrophages. (k) Six2[+] populations are extended in MΦ-depleted embryos compared to controls (two-tailed t-test; p=0.0031). (l) Rostral Six2[+] clusters are larger in area in MΦ-depleted embryos compared to controls (two-tailed Mann-Whitney test; p=0.0042). Scale bars = 100 μm.

DOI: https://doi.org/10.7554/eLife.43271.003

*Figure 1 continued on next page*

*Figure 1 continued*

The following figure supplements are available for figure 1:

**Figure supplement 1.** E10.5 immunostaining.

DOI: https://doi.org/10.7554/eLife.43271.004

**Figure supplement 2.** Macrophage depletion system and its consequences.

DOI: https://doi.org/10.7554/eLife.43271.005

**Figure supplement 3.** Rostral nephrogenic progenitor cell engulfment by macrophages.

DOI: https://doi.org/10.7554/eLife.43271.006

**Figure supplement 4.** Ureteric bud development is delayed in E11.5-E12.5 macrophage-depleted embryos.

DOI: https://doi.org/10.7554/eLife.43271.007

## Kidneys are simultaneously colonised by macrophages and vascularised

We next sought to characterise macrophage arrival and localisation in the early kidney after ureteric bud outgrowth. The ureteric bud had invaded the metanephric mesenchyme by E11 and bifurcated by E11.5 (*Figure 2a–b*). At E11-E11.5, numerous F4/80$^+$ macrophages and blood vessels were present in the peri-Wolffian mesenchyme (situated between the metanephric mesenchyme and Wolffian duct), but relatively few were present within the metanephric mesenchyme (*Figure 2a–c*). The few F4/80$^+$ macrophages within the metanephric mesenchyme were most often at its border nearest the peri-Wolffian mesenchyme (*Figure 2b*), in agreement with macrophage localisation in *Csf1r*$^{EGFP}$ kidneys (*Figure 2—figure supplement 1*) (*Rae et al., 2007*).

Yolk sac-derived macrophage progenitors are trafficked into the embryo-proper via the bloodstream (*Stremmel et al., 2018*). Upon reaching the embryo-proper, these primitive macrophages exit the vascular system and can invade newly forming organs, such as the brain, via extravascular tissue migration (*Cuadros et al., 1993*; *Fantin et al., 2010*; *Herbomel et al., 2001*). Up until E12.5, kidney macrophages are yolk sac-derived (*Hoeffel et al., 2015*). Whole-mount immunostaining demonstrated that F4/80$^+$ macrophages within the E11-E11.5 peri-Wolffian mesenchyme were always extravascular (*Figure 2a–b*), which led us to hypothesise that these macrophages travel via trans-tissue migration. Indeed, time-lapse imaging of cultured E11.5 *Csf1r*$^{EGFP}$ kidneys indicated that these macrophages migrate via extravascular routes and regularly interact with other macrophages as well as the abluminal surfaces of isolectin B$_4$-labelled blood vessels (*Video 3*).

Blood vessels begin entering the kidney from the peri-Wolffian mesenchyme between E11.5-E12 when interstitial tissue regions first form (*Munro et al., 2017b*). We hypothesised that macrophages follow similar routes (in time and space) to colonise the kidney. To examine this, we immunostained and imaged whole-mount kidneys at relevant time points. At E11.5, Six2$^+$ nephron progenitor cells were present throughout the metanephric mesenchyme as a single population, which contained very few macrophages (*Figure 2d*). By E12.5, the Six2$^+$ population had split (*Figure 2e*), resulting in the 'capping' of each ureteric bud tip (*Herring, 1900*; *Short et al., 2014*). As Six2$^+$ nephron progenitor populations split, interstitium forms in the fissures and becomes vascularised (*Airik et al., 2006*; *Daniel et al., 2018*; *Munro et al., 2017b*). Whole-mount imaging of E12.5 kidneys demonstrated that F4/80$^+$ macrophages had occupied these interstitial fissures and commonly wrapped around the interstitial blood vessels (*Figure 2e–f*).

Collectively, these data suggest that macrophages and blood vessels regularly interact as they enter the early kidney at the same time. Static images of whole-mount kidneys indicate that this occupation occurs between E11.5 and E12.5 as the nephron progenitor population splits (*Figure 2g*). At this stage, blood vessels and most macrophages reside within the renal interstitium, rather than the 'cap mesenchyme'

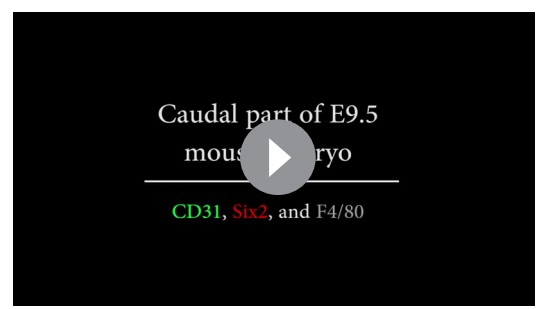

**Video 1.** E9.5 3D caudal-part mouse embryo (Six2, F4/80, and CD31). Few F4/80$^+$ macrophages are present in the embryo-proper at this stage.

DOI: https://doi.org/10.7554/eLife.43271.008

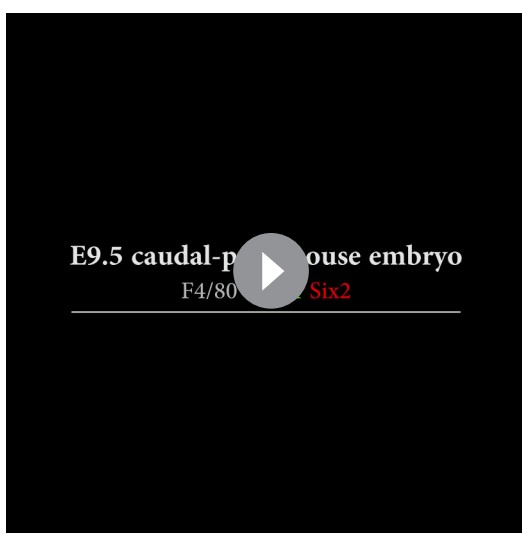

**Video 2.** E9.5 and E10.5 caudal-part mouse embryos z-plane scroll (Six2, F4/80, CD31, and erythrocytes [heme autofluorescence]).
DOI: https://doi.org/10.7554/eLife.43271.009

regions, which are rich in Six2$^+$ nephron progenitor cells.

## Most kidney macrophages in the nephrogenic zone are perivascular throughout foetal development

We next investigated whether the close association between macrophages and endothelial cells continued throughout later kidney development. First, we confirmed that F4/80 specifically marked myeloid cells in the developing kidney by co-staining *Csf1r$^{EGFP}$* kidneys with anti-GFP and anti-F4/80. At E14.5, all detectable F4/80$^+$ cells co-expressed Csf1r (100 ± 0%, mean ±SEM) (*Figure 3a*; *Video 4*). We then immunostained and optically cleared E13.5, E15.5, and E18.5 kidneys to explore organ-wide macrophage localisation. F4/80, CD31, and Gata3 co-staining demonstrated that macrophages were present throughout the medullary and cortical portions of developing kidneys and were often localised around the renal vasculature (*Figure 3—figure supplement 1*; *Video 5*). The type of association of macrophages with blood vessels depended on whether those vessels were anatomically large- or small-calibre. Macrophages often arranged parallel to major renal blood vessels, such as segmental arteries, but did not typically interact with their endothelial cells (*Figure 3b*; *Figure 3—figure supplement 1*). In contrast, macrophages directly interacted with endothelium of small calibre vessels, such as the newly forming cortical nephrogenic zone blood vessels (*Figure 3c–d*).

Important organogenetic processes occur in cycles within the cortical nephrogenic zone of the developing kidney: ureteric bud tips branch, nephron progenitor populations split, nephrogenesis initiates, and interstitial vascular plexuses form (*Herring, 1900*; *Lindström et al., 2018b*; *Munro et al., 2017b*; *Short et al., 2014*). Due to the significance of this zone in renal organogenesis, we next focused on macrophages within this region. Using the E17.5 peripheral nephrogenic zone as a representative example, we determined that 89.6 ± 1.45% (mean ±SEM) of macrophages localised within the interstitium, the remainder being within the cap mesenchyme (nephron progenitor populations; *Figure 3e*; *Figure 3—figure supplement 2*). Of the interstitial macrophages, 78.5 ± 2.1% (mean ±SEM) were in contact with the abluminal surface of blood vessels, and we defined these cells as perivascular macrophages (*Figure 3e*; *Figure 3—figure supplement 2*). The alignment of perivascular macrophages, which we defined based on their longest axis, strongly correlated with the alignment of their associated blood vessel (*Figure 3f*). This tendency of macrophages to localise around the nephrogenic zone vasculature was qualitatively consistent from E13.5 and throughout prenatal kidney development (*Figure 3—figure supplement 3*).

Nephrogenic zone blood vessels are at a vascular front in the developing kidney (i.e. a site of neovascularisation; *Munro et al., 2017b*). In this region, the extent of macrophage interaction with the vasculature and vascular basement membrane varied: a single macrophage may have peri-, trans-, extra-, and intra-vascular regions (*Figure 3g–g'*). We did not detect any entirely intravascular F4/80$^{high}$ kidney macrophages, indicating that these cells either arrived via extravascular routes, that precursor cells gained F4/80$^{high}$ status only after exiting the vasculature, or a combination thereof. Although most macrophages were interstitial, they never expressed the renal cortical interstitial markers Meis1/2 (*Brunskill et al., 2008*) (*Figure 3h*), consistent with the documented exogenous origins of kidney macrophages (*Hoeffel et al., 2015*; *Kobayashi et al., 2014*).

Within the cortical nephrogenic zone, F4/80$^{high}$ macrophages typically co-stained for the mannose receptor, CD206, a marker associated with perivascular/mature macrophage status (*Figure 3i–j*) (*Faraco et al., 2016*; *Galea et al., 2005*). Even at E11.5, before the nephrogenic zone formed, F4/80$^+$ macrophages typically co-expressed CD206 (*Figure 3i*). Immunostaining confirmed that CD206$^+$

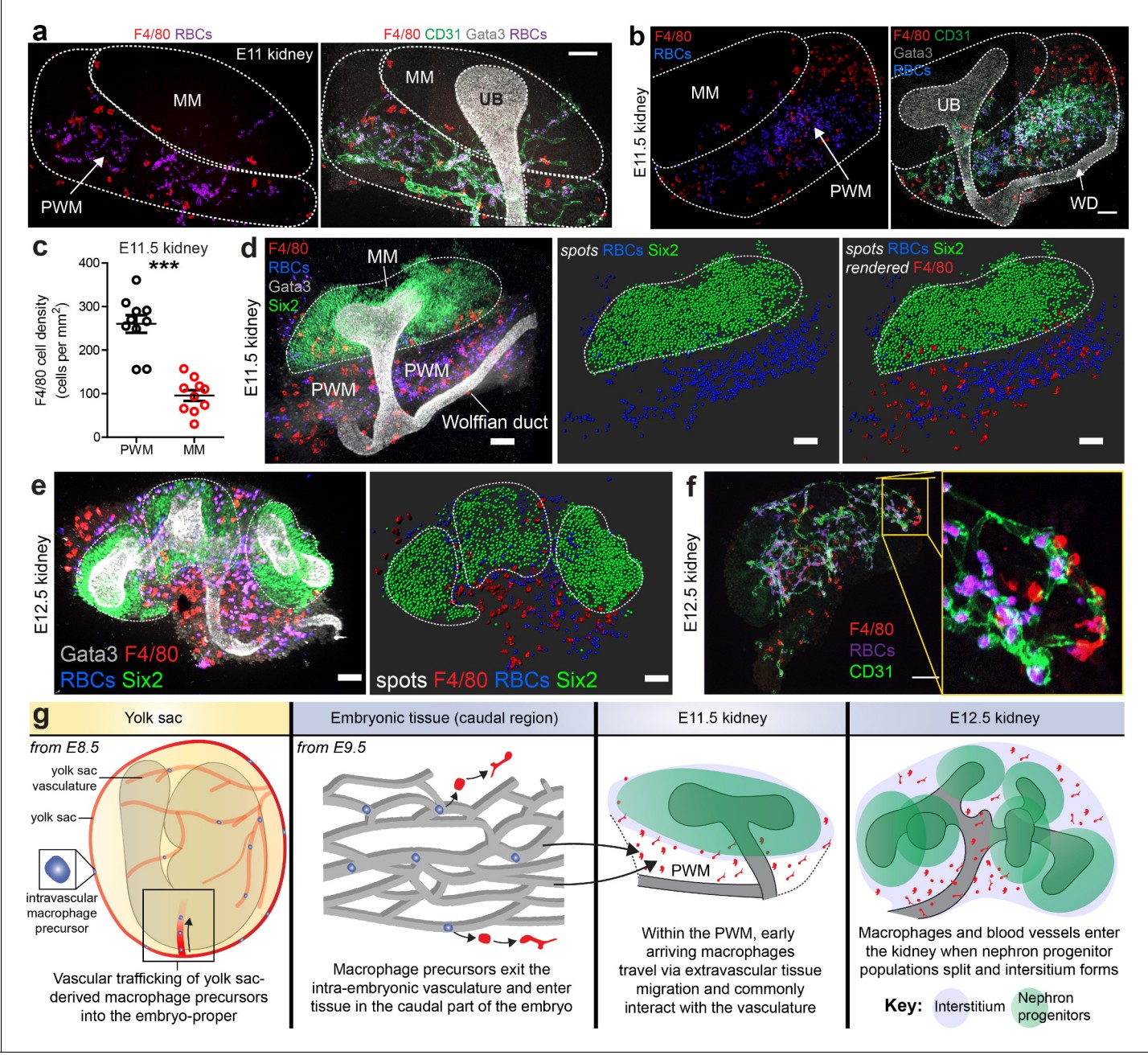

**Figure 2.** Arrival of macrophages in the early kidney. (**a–b**) E11 and E11.5 kidneys: Macrophages localise in high numbers within the vascularised peri-Wolffian mesenchyme (PWM) but not within the metanephric mesenchyme (MM). Staining shows macrophages (F4/80), ureteric bud (UB; Gata3), erythrocytes (heme autofluorescence), and vasculature (CD31) (stitched confocal z-stacks; a = 2×1 tiles, b = 2×2 tiles). (**c**) Macrophage density is higher in the peri-Wolffian mesenchyme than the metanephric mesenchyme (macrophage density in peri-Wolffian mesenchyme (PWM) = 260.20 ± 20.08 per mm$^2$ [mean ±SEM]; macrophage density in metanephric mesenchyme (MM) = 95.86 ± 12.29 per mm$^2$ [mean ±SEM]; two-tailed paired t-test [testing whether densities were the same]; p<0.0001; n = 10 kidneys from five embryos). (**d**) At E11.5, nephron progenitors (Six2) gather as a single population, which is predominantly devoid of macrophages (F4/80) and red blood cells (RBCs; heme autofluorescence). (**e**) By E12.5, nephron progenitor cell populations have split, creating fissures in the mesenchyme that are occupied by macrophages and erythrocytes. The dashed white lines represent the boundaries of Six2$^+$ nephron progenitor populations. Images are 3D z-projections; as a result, macrophages and RBCs appearing within Six2$^+$ populations may in fact localise outside (in 3D). (**f**) At E12.5, many kidney macrophages are perivascular. (**g**) Working model for macrophage arrival in the kidney (based on our results and data from *Stremmel et al., 2018*). Scale bars = 100 μm (except **e**), which = 70 μm).

DOI: https://doi.org/10.7554/eLife.43271.010

The following figure supplement is available for figure 2:

**Figure supplement 1.** E11.5 *Csf1r$^{EGFP}$* kidney.

*Figure 2 continued on next page*

*Figure 2 continued*

DOI: https://doi.org/10.7554/eLife.43271.011

macrophages were often perivascular (*Figure 3—figure supplement 4*) and demonstrated that CD206 predominantly localised in intracellular vesicle membranes, while F4/80 localised in the plasma membrane (most highly at sites of lamellipodia and filopodia) (*Figure 3k–k'*).

A previous study demonstrated a tight relationship between macrophages and nephrons (*Rae et al., 2007*), the latter of which develop and mature deep within the nephrogenic zone (*Figure 3—figure supplement 5*). In accordance with this finding, we observed macrophages in contact with the basement membrane of developing nephrons (*Figure 3—figure supplement 5*; *Video 6*). Macrophages also frequently interacted with blood vessels that wrapped around the developing nephron tubules (*Figure 3—figure supplement 5*; *Video 6*).

These data show that F4/80$^+$CD206$^+$ macrophages in the cortical nephrogenic zone are mainly perivascular throughout the embryonic period of renal organogenesis. These macrophages align with and wrap around newly forming cortical interstitial blood vessels.

## Pro-defence Gal3$^+$ myeloid cells intermingle with pro-development F4/80$^+$CD206$^+$ macrophages in the embryonic kidney

In the developing lung, a subpopulation of galectin3$^+$ (Gal3$^+$) myeloid cells are dispersed alongside yolk sac-derived F4/80$^{high}$ macrophages (*Tan and Krasnow, 2016*). Gal3$^+$ cells colonise the lung from E12.5 onwards and are speculated to be foetal liver-/monocyte-derived macrophages (*Tan and Krasnow, 2016*). We stained *Csf1r$^{EGFP}$* embryos with anti-Gal3 and demonstrated that the developing kidney also contained a subpopulation of Gal3$^+$*Csf1rEGFP$^{EGFP+}$* myeloid cells (*Figure 4a*). Gal3$^+$ cells were largely distinct from CD206$^+$ macrophages, although a small percentage of cells co-expressed both proteins (*Figure 4b–d*). The proportion of myeloid cells that were Gal3$^+$ increased substantially later in development, while the relative proportion of CD206$^+$ cells decreased (*Figure 4d*; *Figure 4—figure supplement 1*).

At E17.5, 14.4% ± 2.2% (mean ±SEM) of Gal3$^+$ myeloid cells in the cortical nephrogenic zone were being carried within blood vessels at the point of fixation (*Figure 4e–f'*; *Figure 4—figure supplement 2*). Gal3$^+$ cells were morphologically spherical in comparison to the irregular morphology of the CD206$^+$ macrophages (*Figure 4g*). In later kidney development, tubular lumens also strongly stained for Gal3 (*Figure 4—figure supplement 1*), in agreement with its expression pattern in the developing human kidney (*Winyard et al., 1997*).

As the localisation and morphology of Gal3$^+$ macrophages differed from F4/80$^+$CD206$^+$ macrophages, we next used single-cell RNA sequencing data to explore the heterogeneity between their transcriptional landscapes. We gathered single-cell RNA sequencing data from E18.5 kidney cells using the Smart-seq2 protocol (*Picelli et al., 2013*). We identified kidney macrophages using principle component analyses (PCA) based on the expression of 48 genes chosen based on relevant literature (including macrophage markers) (*Supplementary file 1*). By performing PCA specifically on these macrophages, distinct clusters of F4/80$^{high}$CD206$^{high}$-Gal3$^{low}$ and Gal3$^{high}$F4/80$^{low}$CD206$^{low}$ macrophage were identified (*Figure 4—figure supplement 3*), substantiating our immunofluorescence observations. The mRNA signatures of F4/80$^{high}$CD206$^{high}$ cells, but not of Gal3$^{high}$ cells, were consistent with their being mature macrophages (signatures based on *Mass et al., 2016*) (*Figure 4h–i*).

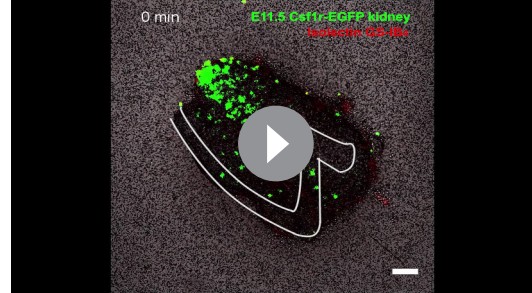

**Video 3.** Time-lapse showing macrophages and isolectin-B4-labelled blood vessels within cultured E11.5 *Csf1r$^{EGFP}$* kidneys. Isolectin-B4 also labels macrophages. White arrows present in the video between 18–40 s show examples of perivascular macrophages.

DOI: https://doi.org/10.7554/eLife.43271.012

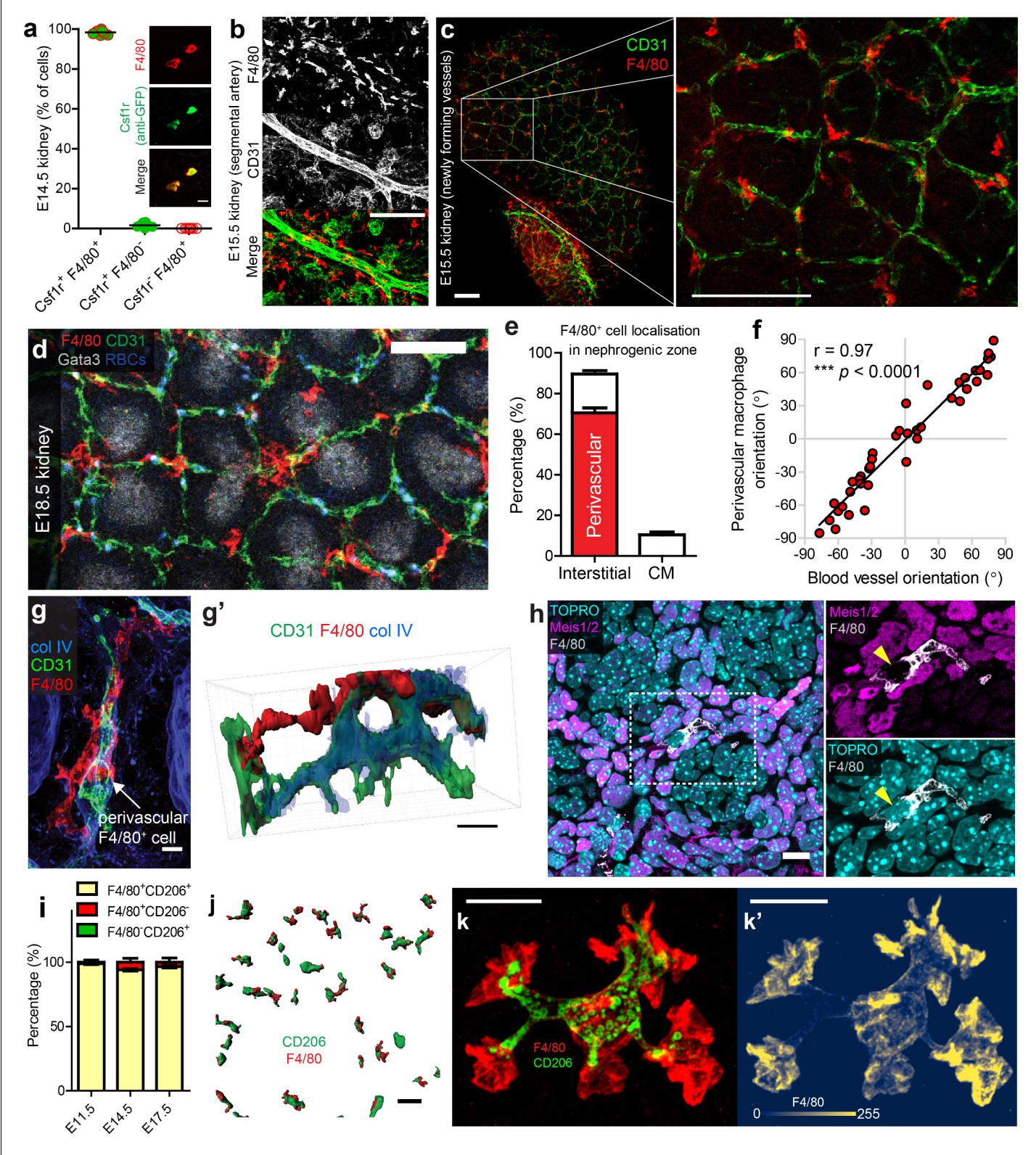

**Figure 3.** Most F4/80[high]CD206[high] kidney macrophages are interstitial perivascular cells. (**a**) All F4/80[+] cells co-express Csf1r (anti-GFP[+]) in the developing kidney (n = 730 cells from 8 z-slices of a cleared E14.5 'MacGreen' kidney). Inset boxes show example Csf1r[+]F4/80[+] cells. (**b**) Macrophages arrange parallel to major vessels in the E15.5 kidney, but rarely contact their endothelia. (**c–d**) Macrophages interact with newly forming vessels in the cortical nephrogenic zone (**c** = E15.5 and **d** = E18.5). (**e**) Most kidney macrophages in the E17.5 nephrogenic zone are interstitial perivascular cells

*Figure 3 continued on next page*

Figure 3 continued

(n = 10 z-stacks from four kidneys). The red bar shows the percentage of interstitial macrophages that are perivascular. CM, cap mesenchyme. (**f**) Macrophages align with their underlying blood vessels (p for correlation <0.0001; n = 40 macrophage/blood vessel pairs from n = 8 kidneys at E17.5). (**g–g'**) Example of a perivascular macrophage in the nephrogenic zone, with intra-, peri-, and extra-vascular regions (**g'** is a 3D rendered version of **g**). (**h**) Macrophages do not express the renal interstitial cell markers, Meis1/2. (**i**) Percentage of cells singly or doubly positive for CD206 and F4/80 in the E11.5 kidney and the cortical nephrogenic zone of E14.5 and E17.5 kidneys. Discernible co-staining at E11.5, 98.8 ± 0.71 [mean ±SEM], n = 5 kidneys; E14.5, 94.1 ± 1.0% of cells, n = 10; at E17.5, 96.5 ± 1.0% of cells, n = 10. (**j**) Representative image of CD206$^+$F4/80$^+$ cells (3D rendered) in nephrogenic zone. (**k–k'**) High-resolution 3D image of an F4/80$^+$CD206$^+$ macrophage. (**k'**) shows cellular intensity of F4/80 using colourmap developed by *Nuñez et al. (2018)*. Scale bars: b-c = 100 µm; (**d**) **j** = 50 µm; **g** = 5 µm; **g'**) =, h, k-k'=10 µm.

DOI: https://doi.org/10.7554/eLife.43271.013

The following figure supplements are available for figure 3:

**Figure supplement 1.** Macrophage arrangement relative to major renal blood vessels.
DOI: https://doi.org/10.7554/eLife.43271.014

**Figure supplement 2.** F4/80$^+$macrophages predominantly avoid nephron progenitor populations in the nephrogenic zone.
DOI: https://doi.org/10.7554/eLife.43271.015

**Figure supplement 3.** Most F4/80$^+$macrophages localise around the interstitial vasculature in the nephrogenic zone.
DOI: https://doi.org/10.7554/eLife.43271.016

**Figure supplement 4.** CD206$^+$macrophages in the nephrogenic zone.
DOI: https://doi.org/10.7554/eLife.43271.017

**Figure supplement 5.** Macrophages interact with nephrons and their vasculature in the deep nephrogenic zone.
DOI: https://doi.org/10.7554/eLife.43271.018

PANTHER gene over-representation testing (http://www.pantherdb.org/) demonstrated that genes associated with biological processes such as immune response (false discovery rate [FDR] corrected p=1.22×10$^{-4}$) and defence response (FDR corrected p=1.25×10$^{-4}$) were enriched in the top 1% of genes expressed by Gal3$^{high}$ cells (*Figure 4j*; *Supplementary file 2*), suggesting that Gal3$^{high}$ macrophages are primed for pathogenic invasion. The top 1% of genes expressed by F4/80$^{high}$-CD206$^{high}$ cells were over-represented in biological processes such as pinocytosis (FDR corrected p=3.8×10$^{-3}$), endocytosis (FDR corrected p=1.6×10$^{-8}$), endosomal transport (FDR corrected p=2.1×10$^{-3}$), and phagocytosis (FDR corrected p=1×10$^{-2}$) (*Figure 4k*; *Supplementary file 3*). The top 1% of genes expressed in both F4/80$^{high}$CD206$^{high}$ cells and Gal3$^{high}$ cells were also over-represented for several cellular components of the endosomal-lysosomal system, consistent with their roles as professional phagocytes (*Supplementary file 4–5*).

Notably, biological processes relating to tissue development were also enriched in the F4/80$^{high}$CD206$^{high}$ macrophages (*Supplementary file 3*). Of these processes, several related to vascularisation, such as vasculature development (FDR corrected p=2.7×10$^{-2}$) and blood vessel morphogenesis (FDR corrected p=3.9×10$^{-2}$). Gene expression by Gal3$^{high}$ cells was not associated with development-related processes (*Supplementary file 2*). These results suggest that F4/80$^{high}$CD206$^{high}$ macrophages may act as pro-developmental cells during kidney development, consistent with a previous report (*Rae et al., 2007*).

We next explored whether distinct populations of CD206$^+$ and Gal3$^+$ immune cells were also present in the developing human kidney by analysing publicly available single-cell RNA sequencing data on GUDMAP (gestational week 14–18; *Lindström et al., 2018a*). In agreement with our results in the mouse, CD206 (MRC1 in the human) and Gal3 (LGALS3 in the human)

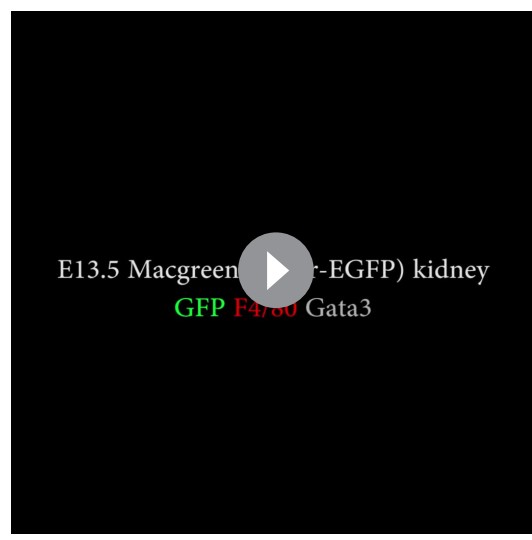

**Video 4.** E13.5 and E14.5 MacGreen kidneys (F4/80, GFP, and Gata3).
DOI: https://doi.org/10.7554/eLife.43271.019

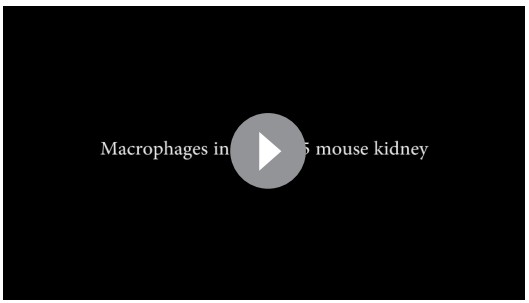

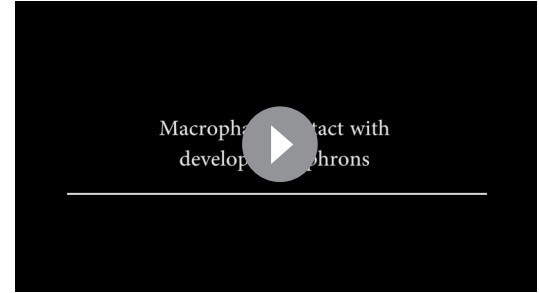

**Video 5.** E15.5 and E18.5 kidney macrophages (F4/80, Gata3, erythrocytes, and CD31). DOI: https://doi.org/10.7554/eLife.43271.020

**Video 6.** Macrophage contact with epithelial tubules of developing nephrons (F4/80, Jagged 1, Laminin). DOI: https://doi.org/10.7554/eLife.43271.021

were differentially expressed in immune cell types of the human embryonic kidney (*Figure 5a–i*). 90% of MRC1[+] cells were macrophages (group 11) (*Figure 5h*), whereas LGALS3[+] immune cells were distributed between the macrophage cluster (group 11; 58% of LGALS3[+] immune cells) and a distinct leukocyte cluster (group 10; 42% of LGALS3[+] immune cells) (*Figure 5i*). The group 10 immune cells specifically expressed L-selectin (SELL) (*Figure 5g*), which functions to tether leukocytes to endothelium (*Stein et al., 1999*). F4/80 (EMR1 in the human) was not expressed by macrophages in the foetal human kidney (*Figure 5j*), consistent with a previous report showing EMR1 is not expressed by human macrophages (*Hamann et al., 2007*).

Collectively, these results show that the developing kidneys of the mouse and human contain heterogeneous populations of immune cells. In the developing mouse kidney, Gal3[high] myeloid cells often travel via the renal vasculature and intermingle with mature F4/80[high]CD206[high] macrophages. Both Gal3[high] and F4/80[high]CD206[high] cells were enriched for mRNAs associated with professional phagocytes, but only F4/80[high]CD206[high] cells were enriched for mRNAs associated with vascular development.

## Macrophages phagocytose components of the renal vascular system and promote endothelial cross-connections

Given that many macrophages in the kidney interact with the interstitial vasculature, we hypothesised that macrophage localisation in the interstitium depended on the presence of blood vessels. To test this, we pharmacologically inhibited vascular development in cultured E12.5 kidneys (using three different pan-VEGF inhibitors: sunitinib, vatalanib, and semaxanib) and compared macrophage localisation. Vascular density was markedly reduced in the treated kidneys (*Figure 6a–c*), but macrophage localisation (in/out of the interstitium) did not differ between groups (one-way ANOVA; p=0.7142; n = 4–9 kidneys per group) (*Figure 6d*). Notably, macrophage density was reduced in vascular-depleted kidneys (*Figure 6e*) and was positively correlated with vascular density when assessing all groups (r = 0.71; p<0.0001) (*Figure 6f*) and even when assessing only control groups (r = 0.74; p=0.004) (*Figure 6g*). Vascular-depletion did not alter other aspects of kidney development (*Figure 6—figure supplement 1*) and an assistant counting blind-coded samples verified the macrophage density count differences between groups (*Figure 6—figure supplement 1*). These data suggest that endothelial-derived signals are not required for macrophage navigation in the renal interstitium.

Renal macrophages engulf cells in the developing mouse and human kidney (*Camp and Martin, 1996*; *Erdosova et al., 2002*); however, the identities of the phagocytosed cells are unknown. We observed F4/80[+] macrophages engulfing erythrocytes and dying endothelial cells in the kidney (*Figure 7a–b*; *Video 7*), demonstrating that, as well as spatially associating with the vasculature, kidney macrophages directly interact with blood and vascular components during development.

Based on kidney macrophage localisation and gene expression (*Figures 1–2*; *Video 5*; *Figure 7c–d*), as well as their known abilities in regulating vascular formation (*Picelli et al., 2013*; *Fantin et al., 2010*; *Rymo et al., 2011*), we hypothesised that macrophages promote renal vascularisation. To test whether macrophages are required for the renal vasculature to develop normally, we

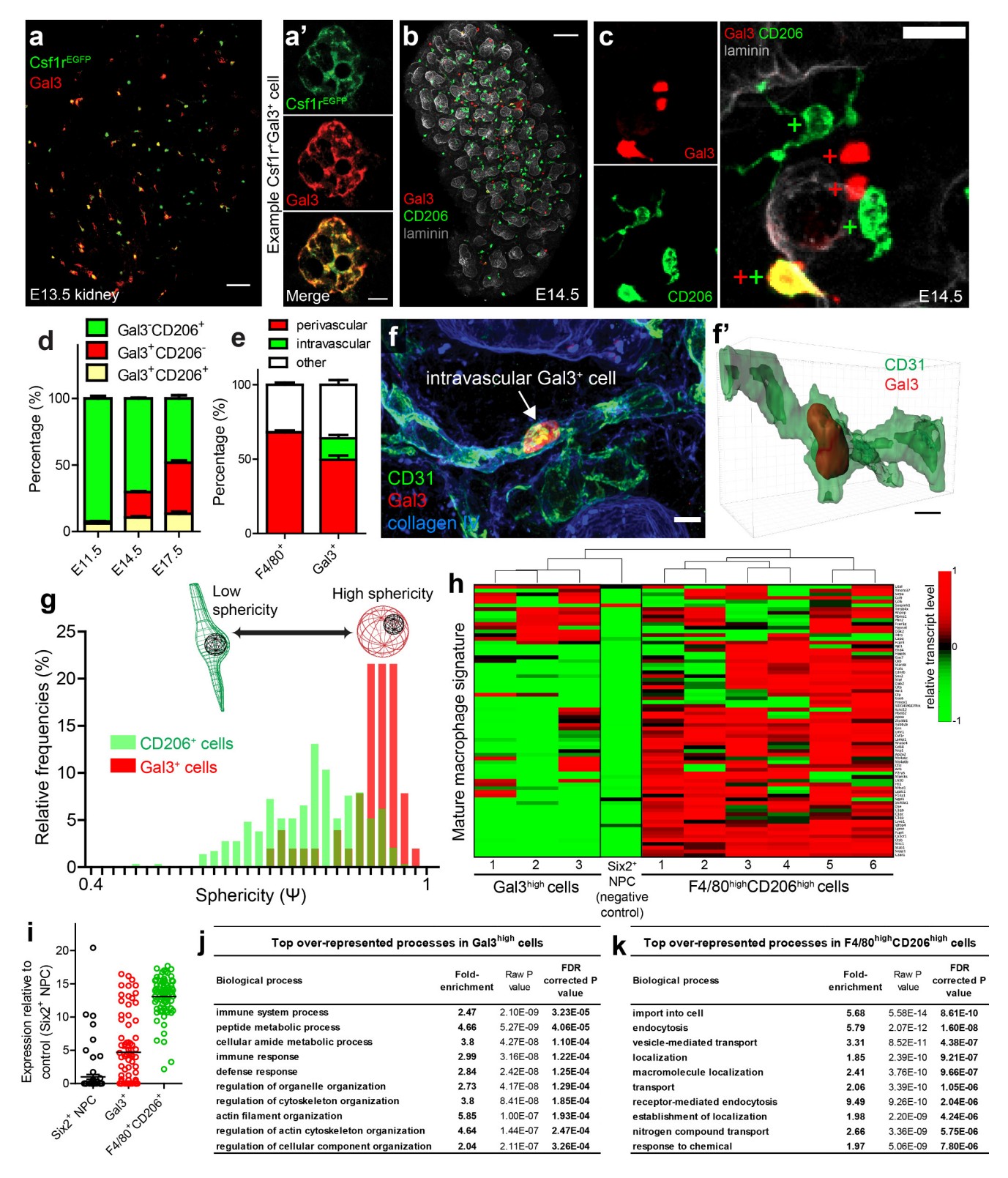

**Figure 4.** Characterisation of Gal3+cells in kidney development. (**a–a'**) Gal3 is expressed by a subset of *Csf1r*EGFP+ cells. (**a'**) shows an example Gal3+Csf1r+ cell. (**b**) Gal3+ and CD206+ cells in the E14.5 kidney. (**c**) Examples of Gal3+CD206-, Gal3-CD206+, and Gal3+CD206+ cells in the E14.5 kidney. (**d**) The proportion of Gal3+ myeloid cells increases over developmental time, while the relative proportion of CD206+ cells decreases. (**e**) Quantification of Gal3+ and F4/80+ macrophage localisation in E17.5 nephrogenic zone (n = 14 fields of view for F4/80 localisation and n = 15 fields of *Figure 4 continued on next page*

*Figure 4 continued*

view for Gal3 localisation). The only intravascular cells were Gal3$^+$. (**f–f'**) Example of an intravascular Gal3$^+$ cell at E17.5. 3D rendering of the intravascular Gal3$^+$ cell from **f**). (**g**) Gal3$^+$ cells are more spherical than CD206$^+$ cells (p<0.0001; two-tailed t-test; n = 290 CD206$^+$ cells; n = 51 Gal3$^+$ cells from E14.5 nephrogenic zone). (**h**) Expression of mature macrophage signature genes by individual F4/80$^{high}$CD206$^{high}$ cells, Gal3$^{high}$ cells, and a nephron progenitor cell (negative control). (**i**) Average mRNA transcript levels for genes associated with mature macrophage status (n = 6 F4/80$^{high}$CD206$^{high}$ cells; n = 3 Gal3$^{high}$ cells; n = 1 Six2$^{high}$ nephron progenitor cell [NPC]). Each data point represents the average expression of a single relevant gene. (**j**) Top 10 over-represented biological processes in Gal3$^{high}$ cells. (**k**) Top 10 over-represented biological processes in F4/80$^{high}$CD206$^{high}$ cells. Scale bars: a = 10 µm; a' and f = 5 µm; b = 100 µm; c = 20 µm.

DOI: https://doi.org/10.7554/eLife.43271.022

The following figure supplements are available for figure 4:

**Figure supplement 1.** Gal3 expression in the developing kidney.

DOI: https://doi.org/10.7554/eLife.43271.023

**Figure supplement 2.** Examples of Gal3$^+$cells being carried within blood vessels of the developing kidney.

DOI: https://doi.org/10.7554/eLife.43271.024

**Figure supplement 3.** Macrophage single-cell RNA sequencing gene expression signatures.

DOI: https://doi.org/10.7554/eLife.43271.025

depleted macrophages in cultured kidney explants using an anti-Csf1r mAb blocking antibody (M279). Csf1r is essential for macrophage proliferation, migration, and survival (*Dai et al., 2002*; *Mouchemore and Pixley, 2012*). E12.5 kidneys were treated for 72 hr with either 20 µg/ml of anti-Csf1r or anti-rat IgG as a control, and macrophages were robustly depleted (*Figure 7e–g*). Consistent with previous studies, anti-Csf1r treated kidneys were smaller than controls (p=0.0046; *Figure 7h*; *Alikhan et al., 2011*; *Rae et al., 2007*; *Sauter et al., 2014*). The area covered by CD31$^+$ endothelium per field of view did not significantly differ between groups (p=0.077; *Figure 7i*); however, macrophage-depleted kidneys had higher numbers of isolated, unconnected endothelial structures (p=0.011; *Figure 7j*) and the average size of CD31$^+$ structures were reduced relative to controls (p=0.046; *Figure 7k*). These data demonstrate that macrophages support production and/or maintenance of endothelial cross-connections in the developing kidney.

## Discussion

In this report, we characterised macrophage arrival, localisation, and heterogeneity in the developing mouse kidney and show that macrophages support the assembly of the kidney and its vasculature. Kidneys depleted of macrophages contained discontinuous endothelial structures, while control-treated kidneys had characteristic net-like vascular plexuses. These data are consistent with macrophages having the capacity to promote vascular anastomoses, as has been described in other biological settings (*Fantin et al., 2010*; *Liu et al., 2016*). Blood vessels provide developing organs with oxygen and essential nutrients to promote growth (*Chung and Ferrara, 2011*). Studies have linked foetal macrophages with enhanced branching morphogenesis, nephrogenesis, and growth during kidney development (*Alikhan et al., 2011*; *Rae et al., 2007*; *Sauter et al., 2014*) and our results showing macrophages promote renal vascularisation may partly explain these links.

While we demonstrate a close relationship between macrophages and developing renal blood vessels, important questions remain to be addressed. Future studies should examine the specific mechanisms by which macrophages mediate kidney vascularisation and should explore other potential roles for renal perivascular macrophages in the embryo. In the adult kidney, perivascular macrophages survey vascular transport for small immune complexes (*Stamatiades et al., 2016*). As the embryo is thought to be relatively sterile (*Tissier, 1900*), there are likely few immune complexes transported via the vasculature during development, but perivascular macrophages may nevertheless survey the bloodstream for factors such as maternal-derived antibodies that have crossed the placental barrier (*Morphis and Gitlin, 1970*). Further, we demonstrated that macrophages can engulf erythrocytes in the developing kidney, suggesting a possible role for renal perivascular macrophages in clearing cells carried via the vasculature.

One way that macrophages shape developing tissues is via their roles in cell clearance (*Wood and Martin, 2017*); for example, macrophages clear excess neurons in the developing brain and spinal cord (*Cuadros et al., 1993*) and engulf dying cells in embryonic footplates to facilitate

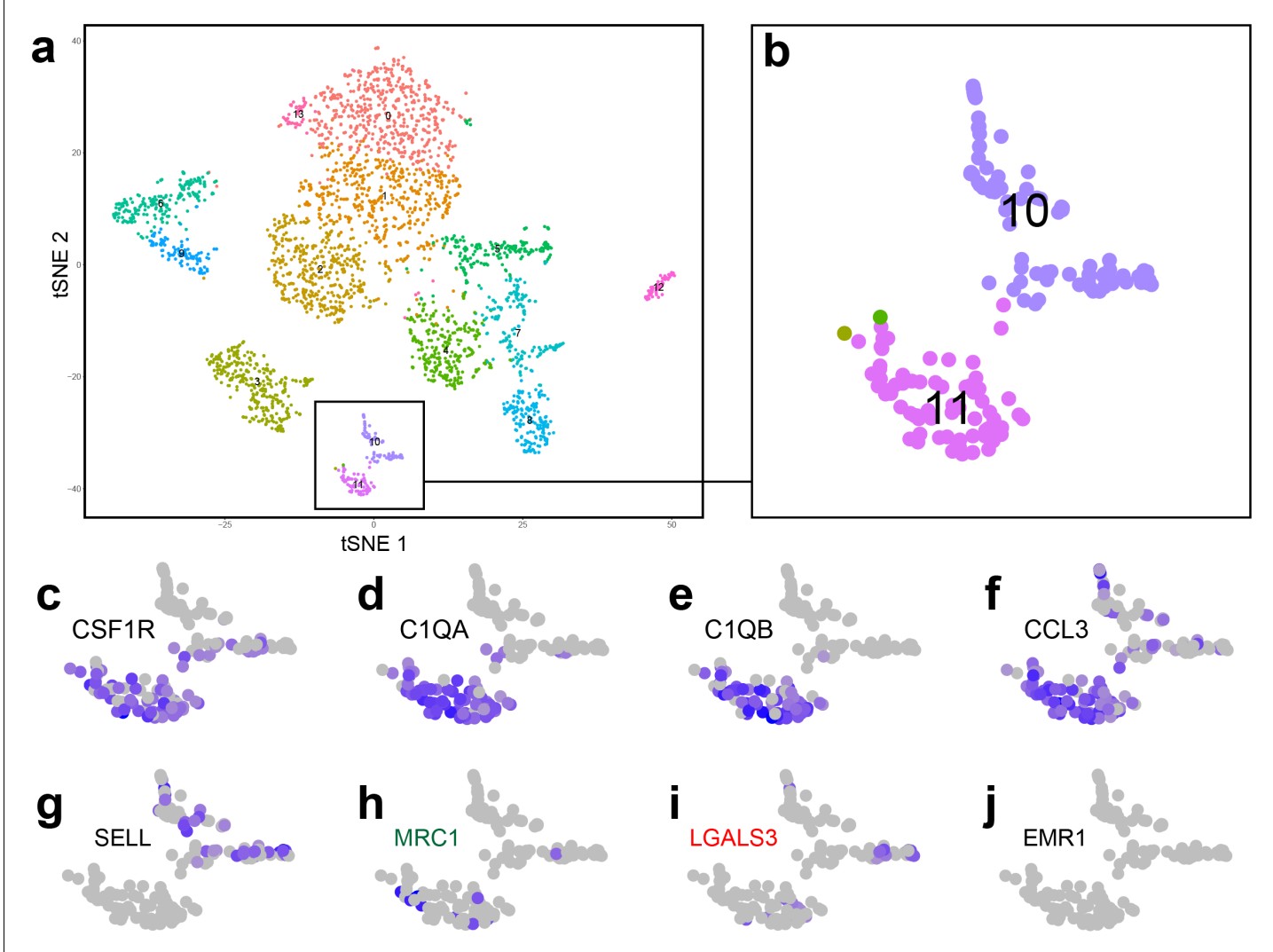

**Figure 5.** Analysis of single-cell RNA sequencing data from human foetal kidney cells. (**a**) tSNE plot showing individual cells from the cortex of the developing human kidney. (**b**) Two clusters (10 and 11) were identified as immune cells. (**c–f**) Cluster 11 cells are macrophages, which express macrophage markers such as (**c**) CSF1R, Colony stimulating factor one receptor; (**d**) C1QA, Complement C1q subcomponent subunit A; (**e**) C1QB, Complement C1q subcomponent subunit B; (**f**) CCL3, macrophage inflammatory protein-1α. (**g**) Cluster 10 cells are a distinct leukocyte cell type, which specifically express L-selectin (SELL). (**h–i**) MRC1+ cells are mostly found within cluster 11 (macrophage group), whereas LGALS3+ cells are distributed within clusters 10 and 11. (**j**) EMR1, the human homolog of F4/80, is not expressed by human kidney macrophages.
DOI: https://doi.org/10.7554/eLife.43271.026

digit formation (*Wood et al., 2000*). Our findings indicate that macrophages play a similar role at the beginning of kidney organogenesis, as they prune the rostral domain of the early nephron progenitor population. Our study failed, however, to expose the precise mechanism(s) by which rostral nephron progenitors are cleared. We observed Six2+ nuclei of rostral nephron progenitors within the cell bodies of some macrophages, suggesting active phagocytosis of these cells, but it is unclear whether other mechanisms also contribute to their clearance; for instance, macrophages may generate signals and/or act as cellular chaperones to promote the caudal migration of these cells.

A trophic role for macrophages in kidney development was first suggested by *Rae et al., 2007*, who demonstrated that treating cultured kidney explants with Csf1, a macrophage mitogen, results in enhanced renal growth and branching morphogenesis associated with increased macrophage numbers. Along with this finding, studies in the postnatal mouse revealed that increased activation of the Csf1/Csf1r pathway results in increased kidney weight and volume (*Alikhan et al., 2011*),

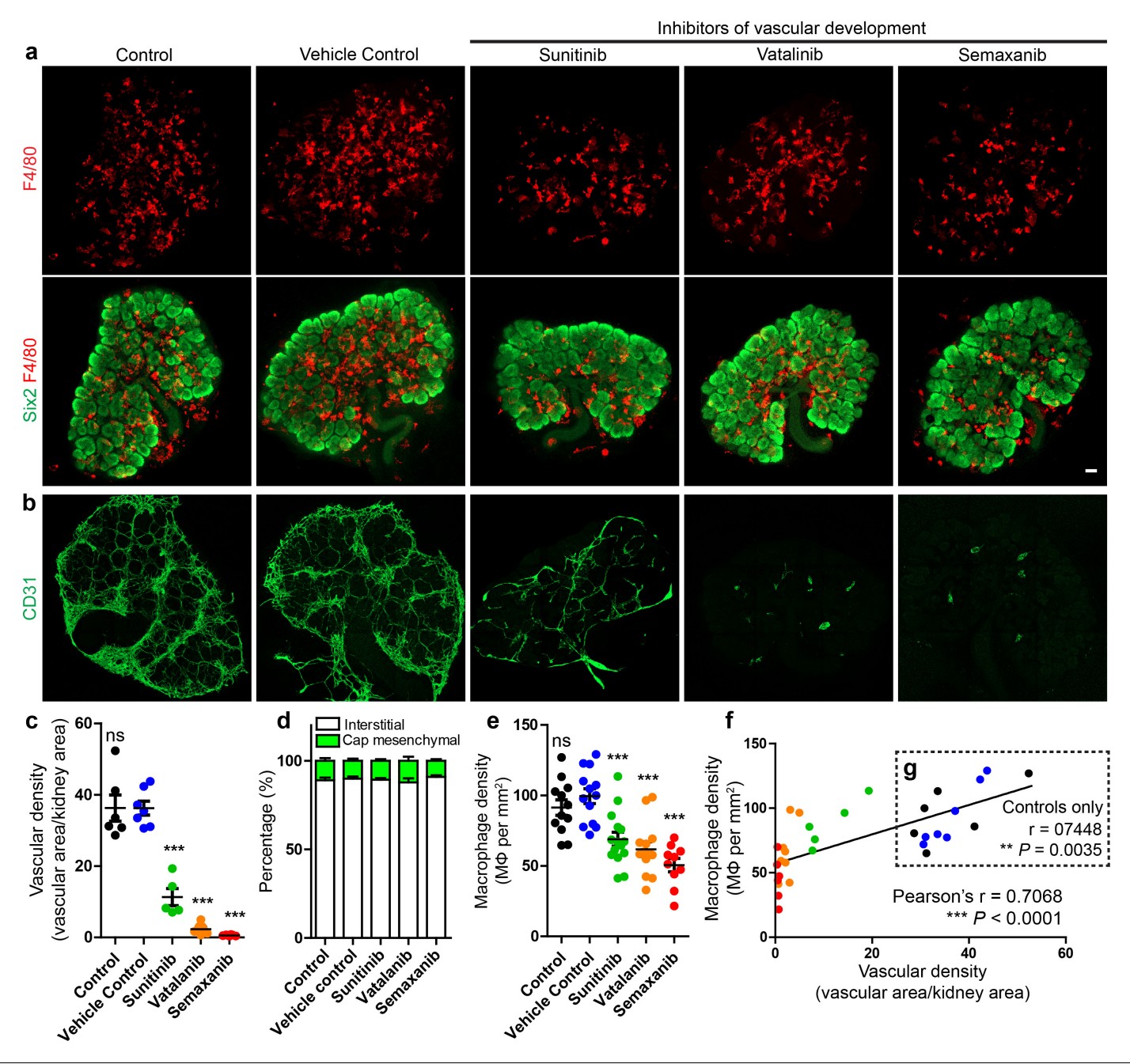

**Figure 6.** Macrophage localisation in the interstitium is not dependent on renal endothelial cells in culture. (**a**) Representative images of macrophages (F4/80) and nephron progenitor populations (Six2) in control and treated kidneys. (**b–c**) Endothelial cells are robustly depleted in sunitinib, vatalinib, and semaxanib treated kidneys (representative images; stitched confocal z-stacks, 3 × 3 tiles; p for overall ANOVA < 0.0001; n = 5–8 per group). (**d**) Macrophage localisation is unaltered between groups (p for overall ANOVA = 0.71; n = 4–9 kidneys per group). (**e**) Macrophage density is decreased in vascular-depleted kidneys (p for overall ANOVA < 0.0001; n = 10–15 kidneys per group). (**f–g**) Macrophage density positively correlated with vascular density in all groups (**f**): r = 0.71, 0.48–0.85 95% CI; p<0.0001; n = 32 kidneys) and in control groups only (**g**): r = 0.74, 0.33–0.92 95% CI; p=0.0035; n = 13 kidneys). In **f–g**, the data point colour represents its group (e.g. green dots represent sunitinib-treated kidneys). Data are from two identical experiments (different antibodies were used for post-fixation sample staining). Macrophage localisation data in **d**) were determined from the first experimental run. Vascular density data used in **c**), (**f**), and **g**) were obtained from the second experimental run. Macrophage density data in **e**) includes data from both experimental runs. Scale bar = 100 µm.

DOI: https://doi.org/10.7554/eLife.43271.027

The following figure supplement is available for figure 6:

*Figure 6 continued on next page*

*Figure 6 continued*

**Figure supplement 1.** Influence of vascular inhibition on other aspects of kidney development.

DOI: https://doi.org/10.7554/eLife.43271.028

whereas blockade of this pathway in the adult results in reduced kidney weight (*Sauter et al., 2014*). Our results add to these findings by showing that, in the absence of macrophages in vivo, the beginning of kidney development is delayed. Collectively, these studies suggest that macrophages act to enhance growth and maturation of renal tissue across the lifespan.

The growth and maturation of renal organoids using current differentiation protocols is limited. Since blood vessels and phagocytic cells are important for renal organogenesis (*Alikhan et al., 2011*; *Rae et al., 2007*; *Sauter et al., 2014*), our results may have significant implications for the generation of kidney organoids. Endothelial cells, monocytes, and macrophages are thought to arrive in the developing kidney from extra-renal sources (*Hoeffel et al., 2015*; *Kobayashi et al., 2014*) and future studies might usefully compare organoid maturation when macrophages and blood vessels are exogenously added to these systems.

In conclusion, we show that foetal kidney macrophages are a multifunctional cell type that frequently interact with newly forming renal blood vessels and encourage organogenesis. These results may inform future macrophage-based strategies for the prevention and treatment of neonatal and adult kidney diseases.

## Materials and methods

### Animals

Wild-type embryonic tissues that were used for descriptive studies and kidney explant culture experiments were obtained from outbred CD-1 mice killed by qualified staff of a UK Home Office–licensed animal house following guidelines set under Schedule 1 of the UK Animals (Scientific Procedures) Act 1986. Experiments were performed in accordance with the institutional guidelines and regulations as set by the University of Edinburgh. The morning of vaginal plug discovery was considered as embryonic day (E) 0.5 and staging was confirmed based on kidney and limb bud morphology.

*Transgenic mice used in this study*: MacGreen ($Csf1r^{EGFP}$) reporter mice have previously been described (*Sasmono et al., 2003*). Transgenic $Csf1rEGFP^{EGFP+}$ embryos were identified based on limb bud GFP fluorescence using a Zeiss Axioscope A1 microscope. $Csf1r^{icre}$ and $Rosa^{DTA}$ mice were bred onto the C57BL/6JOlaHsd genetic background and maintained as heterozygotes. Staging of ~E11.5 $Csf1ricre^{icre+}Rosa^{DTA}$ embryos was based on somite pair (sp) counting and 44–49 sp embryos were used for analyses in *Figure 1*. At E11.5, $Csf1ricre^{icre+}Rosa^{DTA}$ embryos were identified using flow cytometry to identify CD45$^+$ cells and immunofluorescence to identify F4/80$^+$ cells. $Csf1r^{EGFP}$, $Csf1r^{icre}$, and $Rosa^{DTA}$ transgenic mice were maintained at the University of Edinburgh according to locally approved procedures. E18.5 $Six2^{EGFP/Cre}$ mouse embryos were used for mouse single-cell RNA sequencing experiments (for details of this mouse line, see *Kobayashi et al. (2008)*). These mice were maintained at Bar-Ilan University according to locally approved procedures.

### Dissection and organ culture

Kidneys were isolated using previously described methods (*Davies, 2010*). Kidney explants were cultured on sterile 24 mm polyester membrane inserts with 0.4 μm pores (Transwells; Corning 3450) in 1.5 ml of kidney culture medium (KCM; Minimum Essential Medium Eagle [Sigma, M5650] supplemented with 1% penicillin/streptomycin [Sigma, P4333] and 10% foetal calf serum [Invitrogen, 10108165]) per well in 6-well plates. Kidneys were grown at 37°C in a 5% $CO_2$ environment. Kidneys were cultured for indicated times and the medium was changed every 48 hr, unless otherwise stated.

### Whole-mount immunofluorescence

Whole-mount samples were fixed with 4% paraformaldehyde (PFA) for 10–60 mins (depending on sample size). After PFA fixation, samples were washed in 1x phosphate buffered saline (PBS; 1 × 30

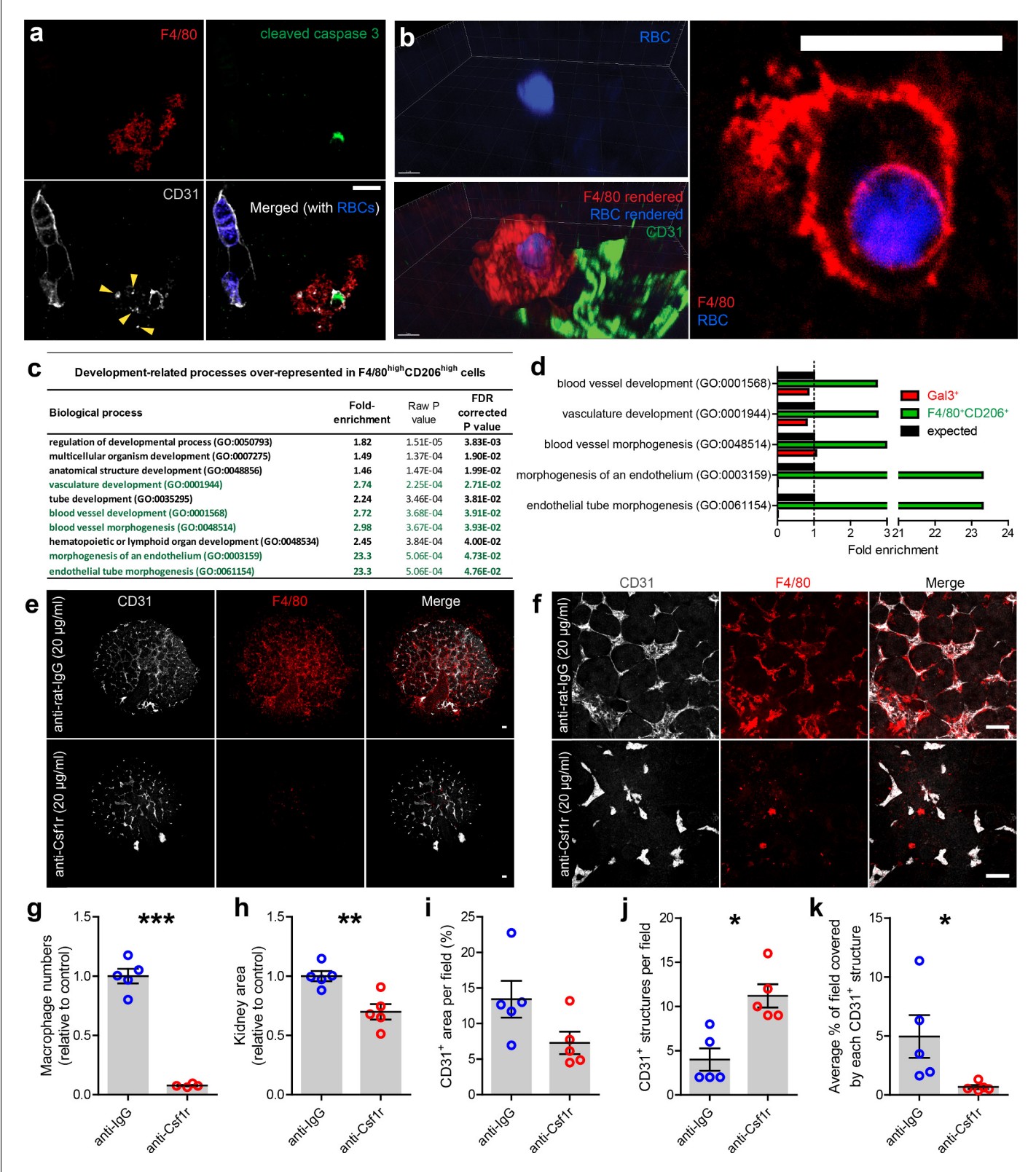

**Figure 7.** Kidney macrophages facilitate vascular development. (**a**) Apoptotic endothelial cell (cleaved caspase 3[+]CD31[+]) in the cortical nephrogenic zone being phagocytosed by an F4/80[+] kidney macrophage. Yellow arrowheads show CD31[+] cell debris within the macrophage cell body. (**b**) F4/80[+] kidney macrophage phagocytosing an erythrocyte (RBC; heme autofluorescence) in the E13.5 kidney. (**c–d**) Genes associated with vascular development are significantly over-represented in the top 1% of expressed genes in F4/80[high]CD206[high] macrophages, but not in the top 1% of the

*Figure 7 continued on next page*

Figure 7 continued

Gal3[high] myeloid cells. Biological processes highlighted in green in a) are all related to vascular development. (e–f) Representative (e) overview images and (f) fields of view of anti-rat IgG (control) and anti-Csf1r (blocking antibody) treated kidney explants. (g–k) Anti-Csf1r treated kidneys had (g) fewer macrophages (p<0.0001), (h) reduced kidney area (p=0.0046), (j) more isolated endothelial structures (p=0.011), and (k) smaller endothelial structures (p=0.046) relative to control treated kidneys (n = 5 biological replicates; 1 field of view per kidney). (i) Vascular density (CD31[+] area per field) did not significantly differ between groups (p=0.077). Data that were normally distributed (Kolmogorov-Smirnov normality testing) were compared using two-tailed t-tests. Only data from j) were not normally distributed and were compared using a non-parametric Mann-Whitney test. Scale bars: a) and b = 10 μm; e) and f = 100 μm.

DOI: https://doi.org/10.7554/eLife.43271.029

mins) and then dehydrated in a methanol: dH$_2$O serial dilution (20%, 40%, 60%, 80%, and then 100%–15 mins per step). After dehydration, samples were either stored at −20°C or directly processed. Samples were rehydrated in a methanol: dH$_2$O serial dilution (80%, 60%, 40%, 20%, and then 0%–15 mins per step). Kidneys were rinsed in PBS (3 × 30 mins) and blocked with 1 x PBS with 5% bovine serum albumin (BSA; Sigma, A9647) and 10% donkey serum (Sigma, D9663) for 1 hr at room temperature or overnight at 4°C. Kidneys were then incubated with primary antibodies, which were diluted in 50% blocking buffer with 50% 1x PBS, overnight at 4°C. Kidneys were rinsed in 1x PBS (3 × 1 hr or overnight at 4°C) and were then incubated with secondary antibodies (in 50% blocking buffer with 50% 1x PBS) overnight at 4°C. For details regarding the antibodies used, see *Supplementary file 6*. Following incubation with secondary antibodies, kidneys were washed in 1x PBS (4 × 1 hr) and mounted onto glass slides in mounting medium (Vectashield; Vectorlabs, H1000). Samples were covered by cover-slips, which were stuck in place on the slide using nail varnish.

## BABB clearing whole-mount immunofluorescence

Samples were optically cleared when deep imaging through thick samples was required. Samples were first fixed in 4% PFA for 1–2 hr. They were washed in 1x PBS (2 × 1 hr) and then dehydrated in a methanol: dH$_2$O serial dilution (20%, 40%, 60%, 80%, and then 100%–15 mins per step). After a 100% methanol wash for 1 hr, samples were incubated with Dent's bleach (methanol: dimethyl sulfoxide [DMSO]: 30% hydrogen peroxide; 4:1:1) for 2 hr, then stored in 100% methanol at −20°C. To continue the protocol, samples were rehydrated in a methanol: dH$_2$O serial dilution (80%, 60%, 40%, 20%, 0%–15 mins per step). They were washed with 1x PBS with 0.2% Triton X-100 (2 × 30 mins). They were then permeabilised for 4 hr using 1x PBS with 0.2% Triton X-100, 300 mM glycine, and 20% DMSO at room temperature. Samples were then blocked with 1x PBS with 0.2% Triton X-100, 3% donkey serum, and 10% DMSO overnight at 4°C. They were then incubated with primary antibodies, which were diluted in 50% blocking buffer with 50% 1x PBS at 4°C for 1–5 days on a rocker. They were then washed in 1x PBS-Tween for 3 × 2 hr. Samples were incubated in secondary antibodies for 24 hr at 4°C diluted in 50% blocking buffer with 50% 1x PBS. Subsequently, samples were washed with 1x PBS-0.2% Tween for 3 × 2 hr and then left in 1x PBS overnight at 4°C. They were dehydrated using a methanol: dH$_2$O serial dilution (15-mins per step; 20%, 50%, 75%, and 100% methanol). They were placed in a glass vial containing 50% benzyl alcohol/benzyl benzoate (BABB) with 50% methanol until they sank to the bottom of the vial and were then cleared in 100% BABB until transparent. After clearing, samples were placed on a slide in a drop of BABB and covered with a glass cover-slip prior to imaging. Erythrocytes were observed in BABB cleared tissue as this clearing method does not remove heme from haemoglobin and because heme acts as a major chromophore under visible light (*Horecker, 1943*; *Munro et al., 2017b*; *Yokomizo et al., 2012*).

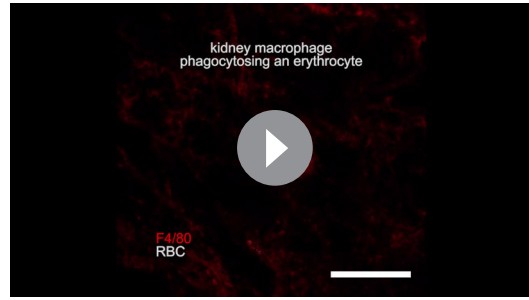

**Video 7.** F4/80[+] macrophages engulfing an erythrocyte (heme autofluorescence) and an apoptotic endothelial cell in kidney development (scroll through individual confocal z-planes).

DOI: https://doi.org/10.7554/eLife.43271.030

### E9.5-E11.5 macrophage density quantifications

At E9.5 and E10.5, numbers of F4/80$^+$ macrophages were manually counted in individual z-planes through cleared caudal regions of mouse embryos using plugin → analyse → cell counter in ImageJ. To calculate macrophage density in the caudal region of embryos, regions devoid of tissue and regions that included nephrogenic cells were traced using the Freehand selections tool on ImageJ and were subtracted from the overall area per field of view. Macrophage density (F4/80 cells per mm$^2$) was calculated by dividing macrophage number by the area of the caudal tissue region in each field of view. Macrophage density within the Six2$^+$ nephrogenic populations were scored in the same way, with the area covered by Six2$^+$ cells on each field of view being calculated using the Freehand selections tool.

To calculate macrophage density along the rostro-caudal axis of the metanephric mesenchyme at E10.5, we used the Straight-line tool on ImageJ to draw a line along the length of the main population of Six2$^+$ cells and measured the length of this line. We divided this line by four to define each quarter of the population along its rostro-caudal axis and counted macrophage numbers per quarter. We then related macrophage numbers (calculated using plugin → analyse → cell counter in ImageJ) to the area of each quarter to calculate macrophage density (F4/80 cells per mm$^2$). We also quantified macrophage density within isolated rostral populations of Six2$^+$ cells at E10.5. Macrophage density was represented on a linear 3-colour scale, with blue indicating the lowest density, black indicating values in the middle, and red indicating the highest density values.

Macrophage density was calculated in the E11.5 peri-Wolffian mesenchyme and metanephric mesenchyme using ImageJ. The metanephric mesenchyme was defined and drawn using the free-hand selection tool by a kidney development researcher blinded to the experimental purpose. To eliminate potential bias based on macrophage or vasculature localisation, only the Gata3 channel (showing ureteric bud) was shown to the blinded researcher. Peri-Wolffian mesenchyme and metanephric mesenchyme areas were calculated and macrophage numbers counted (using plugin → analyse → cell counter in ImageJ) to calculate relevant macrophage densities (F4/80 cells per mm$^2$).

### Characterisation of Six2$^+$ cell populations in E11.5 Csf1r$^{icre}$Rosa$^{DTA}$ embryos

To characterise the total rostro-caudal length of Six2$^+$ populations in macrophage-depleted and control embryos we analysed confocal z-planes of caudal part embryos stained with anti-Six2. The segmented line tool in ImageJ was used to calculate the total length of the Six2$^+$ population from the most caudal to the most rostral Six2$^+$ cell (using analyse → measure in ImageJ). To characterise the area of rostral Six2$^+$ clusters, we used the Freehand selections tool in ImageJ to draw around these cell clusters and used analyse → measure to define the area of these populations.

### Quantifications of co-immunostained cells

*Csf1r and F4/80 co-staining:* Csf1rEGFP$^{EGFP+}$ kidneys were fixed, immunostained, and cleared. An anti-GFP antibody was used to label Csf1rEGFP$^{EGFP+}$ cells. Co-localisation of GFP$^+$ and F4/80$^+$ cells was quantified using plugin → analyse → cell counter in ImageJ. Quantification was based on assessments of 8 z-planes from an E14.5 Csf1rEGFP$^{EGFP+}$ kidney. *F4/80 and CD206 co-staining:* Quantification was performed as described above using antibodies against F4/80 and CD206. Quantifications were made at E11.5, E14.5, and E17.5; at E14.5 and E17.5 quantifications were based on macrophages within the cortical nephrogenic zone. *Gal3 and CD206 co-staining:* Quantification was performed as described above using antibodies against Gal3 and CD206. Quantifications were made at E11.5, E14.5, and E17.5; at E14.5 and E17.5 quantifications were based on cells within the cortical nephrogenic zone.

### Microscopy

All images were generated using the Zeiss LSM800 confocal microscope, except from the time-lapse images, which were generated using the Nikon A1R confocal microscope. Objectives of 5-63x were used. Objective lenses were oil-immersed from 40x upwards. Images were analysed and processed using ImageJ (FIJI) and IMARIS (version 8.3.1).

## Time-lapse imaging

E11.5 Csf1rEGFP$^{EGFP+}$ kidneys were explanted and cultured in Transwells as described above. A 6-well plate containing the Transwells was placed in an imaging chamber and the kidneys were grown at a temperature of 37°C in a 5% $CO_2$ environment. Isolectin-B4 (I32450; ThermoFisher) was added to the culture medium at a concentration of 1:1000. For the video showing the entire kidney, images were taken every 15 mins. For the video showing a small region of the kidney, images were taken every five mins.

## Quantification of F4/80$^+$ macrophage localisation in the nephrogenic zone

Macrophage localisation was quantified in the E17.5 nephrogenic zone by co-staining for F4/80, collagen IV, and Six2. Collagen IV marks blood vessel basement membranes in the developing kidney (*Munro et al., 2017b*). To quantify macrophage localisation, we analysed macrophage location in 3D by scrolling through z-planes. Quantifying macrophage localisation in individual z-planes was necessary because macrophages sitting above the cap mesenchyme in 3D may have appeared within the cap mesenchyme in flattened 2D images - the same was true for macrophage localisation around blood vessels.

## Perivascular macrophage orientation

The alignment of perivascular macrophages was measured by drawing a line along the longest axis of the macrophage using the straight-line tool in ImageJ. The angle of this line was the measured using analyse → measure in ImageJ. The same measurement was performed for the underlying blood vessel, with the straight line being drawn along the blood vessel up to the site of its splitting.

## Macrophage sphericity measurements

The sphericity of Gal3$^+$ and CD206$^+$ cells in the E14.5 kidney was measured using IMARIS (version 8.3.1). Cells were surface-rendered and non-cellular rendered objects manually removed. IMARIS automatically calculated the sphericity of each rendered cell.

## Single-cell RNA sequencing

### Mouse

Single-cell RNA sequencing experiments were performed following the smart-seq2 protocol (*Picelli et al., 2013*). MATLAB and GraphPad were used to prepare single-cell heatmap clustering graphs (using the Bioinformatics Toolbox in MATLAB) and principle component analyses (PCA) graphs (using the Statistics and Machine Learning Toolbox in MATLAB). PCA graphs were prepared based on the expression of 48 genes chosen based on relevant literature. To standardise the expression values for each gene, values were first log-transformed after adding 1. Next, the log gene expression value for each cell was subtracted from the mean log gene expression value of all cells analysed. This value was then divided by the standard deviation, and truncated into the range of −1,1 to eliminate outliers.

### Gene signature analyses

The gene signatures for mature macrophages were based on signatures generated by *Mass et al. (2016)* (available at www.sciencemag.org/content/353/6304/aaf4238/suppl/DC1). Values used for gene signature analyses were based on averaged gene expression values of the different identified cell subtypes.

*Gene set enrichment analyses* - The Panther Gene Overrepresentation Test was used to assess whether the top 1% of expressed genes (234 genes) by Gal3$^+$ and by F4/80$^+$CD206$^+$ cells were enriched for certain biological processes or cellular components (*Mi et al., 2013*; http://pantherdb.org/tools/compareToRefList.jsp). The reference dataset used for this analysis was the entire gene list used in the single-cell analyses. False discovery rate corrected p-values were calculated using Fisher's Exact with FDR multiple test correction.

## Human

Single-cell RNA sequencing data from human foetal kidney cells were accessed from GUDMAP (*Harding et al., 2011*) using the 'Q-Y4GR: Cellular Diversity in Human Nephrogenesis' dataset (*Lindström et al., 2018a*). These data were analysed using Seurat, an R package for single cell genomics (https://satijalab.org/seurat/; *Butler et al., 2018*).

## Macrophage depletion using an anti-Csf1r antibody

E12.5 kidneys were cultured on Transwell filters in 1.5 ml KCM for 3 days with either 20 µg/ml anti-Csf1r mAb blocking antibody (M279) or 20 µg/ml anti-rat IgG (as a control). To retain anti-Csf1r and anti-rat IgG (control) within the medium, KCM was not refreshed during the 3 day culture period. After 3 days, cultured kidneys were directly fixed in methanol and processed for immunofluorescence (as described in the 'Whole-mount immunofluorescence' section, after the fixation step).

Macrophage numbers were quantified using the add spots tool on the F4/80 channel in IMARIS (version 8.3.1). Spot number per kidney was defined as the total macrophage number per kidney. To calculate kidney area, kidneys were drawn using the Freehand selections tool in ImageJ and the area covered by each kidney was measured. To calculate the CD31$^+$ area per field (%), the CD31 channel was prepared for thresholding using process → filters → median [radius = 2 pixels] then image brightness/contrast was adjusted in ImageJ. Images were thresholded using image → adjust → threshold (default thresholding). After thresholding, the percentage of the thresholded area per field of view was measured and defined as the vascular density (% CD31$^+$ area per field). The numbers of isolated CD31$^+$ structures per field were then quantified by a blinded counter that was given code samples using plugin → analyse → cell counter in ImageJ. The number of CD31$^+$ structures per field was divided by the total CD31$^+$ area per field to give the average % of the field covered by each CD31$^+$ structure.

## Vascular growth inhibition

E12.5 kidneys were cultured on Transwell filters for 3 days in either 1.5 ml of KCM-only (control), KCM with vascular development inhibitors, or KCM with DMSO (AppliChem, A3672-0100; vehicle control). Inhibitor concentrations were calculated based on half-maximal inhibitory concentration (IC50) values against Vegfr2 (using values from the IUPHAR/BPS Guide to Pharmacology; http://www.guidetopharmacology.org/). Vatalinib had previously been used at a concentration of 1 µM to inhibit vascular development in cultured kidneys (*Halt et al., 2016*); therefore, we used 1 µM of vatalinib in our experiments. 1 µM of vatalinib is at a concentration that is 47.62 times greater than its IC50 against Vegfr2. To calculate concentrations to use for the other inhibitors, we multiplied sunitinib's and semaxanib's IC50s against Vegfr2 by 47.62 (for consistency). Based on these calculations, the inhibitor concentrations used in the KCM were 1 µM of vatalinib, 1.076 µM of sunitinib, and 9.524 µM of semaxanib. As a vehicle control, DMSO was added at the same volume as the highest volume used for the inhibitors. After 3 days of culture, cultures were stopped, and kidneys were directly fixed in methanol and processed for immunofluorescence.

### Macrophage density quantification

Kidney areas were first measured using the Freehand selections tool in ImageJ. Macrophages and blood vessels were only counted if they fell within the boundary of the kidney (using plugin → analyse → cell counter in ImageJ). To validate the counting method, a second, blinded counter was given coded samples to count macrophage numbers using the same method (n = 3 per group). *Vascular density quantification:* The CD31 channel was separated from other channels in ImageJ. For each sample, the Despeckle and Subtract Background (rolling ball radius = 50) tools were used and the image contrast was enhanced; these steps were performed to increase the accuracy of thresholding based on CD31 signal. Images were thresholded using the Huang method. The kidney area was drawn using the Freehand selections tool in ImageJ, and the percentage area covered by CD31$^+$ structures was taken as the vascular density (% CD31$^+$ area per field). *Macrophage localisation quantification:* To count whether macrophages localised within the interstitium or cap mesenchyme, we stained kidneys with a cap mesenchyme marker, Six2. Macrophages that did not localise within the Six2$^+$ cap mesenchyme were classed as interstitial and the percentages of macrophages in/out of the caps were plotted. *Ureteric bud branching generations quantification:* For each kidney,

we counted the number of ureteric bud bifurcations from the primary bifurcation site (at the upper ureter) to the final bifurcation site (at the kidney's periphery).

## Statistics and data presentation

Data are mean ± standard error of the mean. Data that passed normality testing were analysed using parametric tests. Data that did not pass normality tests were analysed using non-parametric tests (specific tests used are indicated in relevant text). When two experimental groups were being compared, t-tests or the non-parametric equivalent were used. When more than two groups (which were normally distributed) were compared, one-way ANOVAs were performed with post-hoc testing then being used to compare differences between individual groups. All p-values were based on two-tailed comparisons. GraphPad (version 5) was used for statistical testing and graph preparation. IMARIS (version 8.3.1) and Adobe Premiere Pro CC (2015) were used to prepare videos and Adobe Illustrator CC (2015) was used to prepare figures.

# Acknowledgements

We would like to express our gratitude to Karen Chapman, Chris Mills, Clare Pridans, Samanta Mariani, Melanie Lawrence, and Jeremy Hughes for help and/or advice during the preparation of this work. We are also grateful to Anisha Kubasik-Thayil of the IMPACT imaging facility and Martin Waterfall of the Flow Cytometry Core Facility at the University of Edinburgh for their technical assistance. Thanks goes to Alison MacKinnon and Clare Pridans for sharing resources. This work was supported by the Medical Research Council (grant number: MR/K501293/1) and the Biotechnology and Biological Sciences Research Council (BBSRC; grant number BB/P013732/1).

# Additional information

## Funding

| Funder | Grant reference number | Author |
| --- | --- | --- |
| Medical Research Council | MR/K501293/1 | Jamie A Davies |
| Biotechnology and Biological Sciences Research Council | BB/P013732/1 | Peter Hohenstein |

The funders had no role in study design, data collection and interpretation, or the decision to submit the work for publication.

## Author contributions

David AD Munro, Conceptualization, Data curation, Formal analysis, Validation, Investigation, Visualization, Methodology, Writing—original draft, Project administration, Writing—review and editing; Yishay Wineberg, Formal analysis, Investigation, Methodology; Julia Tarnick, Conceptualization, Investigation, Writing—review and editing; Chris S Vink, Investigation, Writing—review and editing; Zhuan Li, Investigation, Methodology; Clare Pridans, Conceptualization, Resources, Methodology, Provided various materials for this study (mice and reagents) as well as experimental ideas; Elaine Dzierzak, Resources, Writing—review and editing; Tomer Kalisky, Resources, Methodology; Peter Hohenstein, Conceptualization, Resources, Supervision, Writing—review and editing; Jamie A Davies, Conceptualization, Resources, Supervision, Funding acquisition, Validation, Writing—original draft, Project administration, Writing—review and editing

## Author ORCIDs

David AD Munro ⓘ http://orcid.org/0000-0002-3521-2121
Clare Pridans ⓘ http://orcid.org/0000-0001-9423-557X
Tomer Kalisky ⓘ http://orcid.org/0000-0003-4733-262X
Jamie A Davies ⓘ http://orcid.org/0000-0001-6660-4032

## Ethics

Animal experimentation: Wild-type embryonic tissues that were used for descriptive studies and kidney explant culture experiments were obtained from outbred CD-1 mice killed by qualified staff of a UK Home Office-licensed animal house following guidelines set under Schedule 1 of the UK Animals (Scientific Procedures) Act 1986. Experiments were performed in accordance with the institutional guidelines and regulations as set by the University of Edinburgh.

## Decision letter and Author response

Decision letter https://doi.org/10.7554/eLife.43271.041
Author response https://doi.org/10.7554/eLife.43271.042

## Additional files

### Supplementary files

• Supplementary file 1. Genes used for single-cell RNA sequencing PCA graphs.
DOI: https://doi.org/10.7554/eLife.43271.031

• Supplementary file 2. Biological processes enriched in the top 1% of genes expressed by Gal3$^{high}$ cells.
DOI: https://doi.org/10.7554/eLife.43271.032

• Supplementary file 3. Biological processes enriched in the top 1% of genes expressed by F4/80$^{high}$ CD206$^{high}$ cells.
DOI: https://doi.org/10.7554/eLife.43271.033

• Supplementary file 4. Cellular components enriched in the top 1% of genes expressed by Gal3$^{high}$ cells.
DOI: https://doi.org/10.7554/eLife.43271.034

• Supplementary file 5. Cellular components enriched in the top 1% of genes expressed by F4/80$^{high}$ CD206$^{high}$ cells.
DOI: https://doi.org/10.7554/eLife.43271.035

• Supplementary file 6. Antibodies used in this study.
DOI: https://doi.org/10.7554/eLife.43271.036

• Transparent reporting form
DOI: https://doi.org/10.7554/eLife.43271.037

### Data availability

Data analysed during this study is included in the supporting files.

The following previously published dataset was used:

| Author(s) | Year | Dataset title | Dataset URL | Database and Identifier |
|---|---|---|---|---|
| Lindström NO, Guo J | 2018 | Q-Y4GR: Cellular Diversity in Human Nephrogenesis | https://gudmap.org/chaise/record/#2/RNA-Seq:Study/RID=Q-Y4GR | GUDMAP, Q-Y4GR |

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
