## [Decision Letter]

Thank you for submitting your article "Macrophages restrict the nephrogenic field and promote endothelial connections during kidney development" for consideration by *eLife*. Your article has been reviewed by three peer reviewers, one of whom is a member of our Board of Reviewing Editors, and the evaluation has been overseen by Didier Stainier as the Senior Editor. The following individual involved in the review of your submission has agreed to reveal her identity: Melissa H Little (Reviewer #2).

The reviewers have discussed the reviews with one another and the Reviewing Editor has drafted this decision to help you prepare a revised submission.

Summary:

The developing murine kidney has previously been shown to contain a resident macrophage population at 12.5dpc, however the specific role of this population and their origin is not well characterized. Previous studies have shown that kidney morphogenesis in vitro is improved in the presence of macrophage colony stimulating factor while differentiation is impeded by loss of this population, suggesting a role for this macrophage population in renal development. In this study, Munro et al. extend these studies, and they propose that there are two waves of migration; an early population of macrophages derived from the yolk sac, and a later arriving population derived from HSCs. They show that depleting macrophages during differentiation using mouse lines expressing DT result in mis-patterning of the Six2^+^ mesenchymal progenitor population, suggesting that macrophages have a role in clearing of this population from more anterior regions to define the metanephric field. Finally, they argue a close spatial localization of the resident macrophage population with the developing vasculature and a disruption to endothelial cell anastomosis and plexus formation upon macrophage depletion, but no converse adverse effect on macrophage number after depletion of endothelial populations.

Specific comments:

1) A key claim of the manuscript is that the position of the macrophages is in close association with the forming vasculature. The imaging performed to make this argument can only place the macrophages with respect to the endothelial and collecting duct populations. What is not shown is the location of the forming nephrons. The data from Rae et al. shows a tight relationship between the resident macrophages and the nephron epithelium itself. Indeed, in support of this as a primary and critical relationship, it has been reported that the epithelium itself can produce CSF1 in response to injury. Hence, a macrophage/nephron epithelial relationship may be just as important. The manuscript argues that the role of these resident macrophages is in vessel morphogenesis alone however their location is equally close to the forming tubules, which should also be shown. The authors should investigate this possibility. There are many different antibodies they could use to localize with respect to tubule. EpCAM and PAX2 would be a good start. This could be done as whole-mount and images in low or high resolution. The analysis of resident macrophages wrapping around tubules should be done as whole-mount.

2) Using DTA transgenic lines, the authors deplete the early macrophage population, and they observe clusters of Six2^+^ cells persisting in the rostral domain in the mutant, supporting the idea that these early arriving macrophages may be important for clearance. The proposal that early macrophages are important for clearance implies that there is active phagocytosis of Six2^+^ cells. In the absence of this macrophage population there is a clear separation between a caudal Six2^+^ population and a more rostral population that is perhaps within the mesonephric field. Is it possible that macrophages generate signals that stimulate caudal migration of Six2^+^ cells rather than clearance? The presence of an increased total number of Six2^+^ cells in the DT experiments would help support the proposal that macrophages clear Six2^+^ cells via phagocytosis.

In addition, it would be nice to know more about the aftereffects of failed clearance of Six2^+^ cells; for example, do ectopic nephrons form in the area corresponding to the mesonephros? Six2 cells secrete GDNF and other molecules, are there ectopic ureteric bud branches as a consequence of persistence of these cells?

3) The proposal that a second wave of macrophages arrive via the vasculature needs to be better supported; for example, with movies, better quality images or alternately, the tone of the text needs to be softened.

Other comments:

1) The study makes a claim that a major role for the resident macrophage population is assisting in the joining of endothelial structures to form a mature and connected vasculature. This claim appears to rest on the quantification presented in Figure 7E-K. It is not clear how the feature measured in Figure 7K was reached. This refers to% of field covered by each CD31^+^ structure. How is a CD31^+^ structure defined here? In the non-depleted population this is a single plexus and it seems difficult to see how individual cells can be separated. The earlier data in Figure 7 showing phagocytosis is also very difficult to see as the critical panels are simply too small and the contrast, particularly on the blue channel, too high to see what is going on.

2) Lineage tracing experiments are required to definitively prove that Gal3^+^ macrophages seen at E14.5 are not just the F480^+^ macrophages that up-regulate Gal3 expression at this stage. Hence, this point should be toned down unless there is additional evidence supporting 2 lineages.

3) The authors find that macrophages are localized between the Six2^+^ nephron progenitor cells and the nephric duct prior to the onset of ureteric bud outgrowth, and by E10.5 were localized in a rostral domain of Six2^+^ cells (this is the regressing mesonephros?) where the authors hypothesize they may be important for clearance.

In terms of the broad community, it would be nice to link macrophage biology in the kidney with what is known in other systems. For example, what signals could be important for migration of early and late macrophage populations during development, are they conserved with pathways that function during pathogenesis or are known regulators likely to be involved? There might be some clues that can be gleaned from the single-cell RNA sequencing study.

---

## [Author Response]

Specific comments:1) A key claim of the manuscript is that the position of the macrophages is in close association with the forming vasculature. The imaging performed to make this argument can only place the macrophages with respect to the endothelial and collecting duct populations. What is not shown is the location of the forming nephrons. The data from Rae et al. shows a tight relationship between the resident macrophages and the nephron epithelium itself. Indeed, in support of this as a primary and critical relationship, it has been reported that the epithelium itself can produce CSF1 in response to injury. Hence, a macrophage/nephron epithelial relationship may be just as important. The manuscript argues that the role of these resident macrophages is in vessel morphogenesis alone however their location is equally close to the forming tubules, which should also be shown. The authors should investigate this possibility. There are many different antibodies they could use to localize with respect to tubule. EpCAM and PAX2 would be a good start. This could be done as whole-mount and images in low or high resolution. The analysis of resident macrophages wrapping around tubules should be done as whole-mount.

We agree that a macrophage/nephron epithelial relationship may be important in kidney development. To address this point, we performed new experiments using whole-mount imaging and we found that macrophages often wrap around developing nephrons, in accordance with the findings of Rae et al., 2007. These data can be found in the new Video (Video 6) and in a new figure supplement (Figure 3—figure supplement 5). These new results are described in paragraph five of subsection “Most kidney macrophages in the nephrogenic zone are perivascular throughout fetal development”.

To aid understanding for non-renal development experts, we have added a cartoon in Figure 3—figure supplement 5 to illustrate the location of nephron formation relative to other cell populations within the nephrogenic zone.

2) Using DTA transgenic lines, the authors deplete the early macrophage population, and they observe clusters of Six2^+^ cells persisting in the rostral domain in the mutant, supporting the idea that these early arriving macrophages may be important for clearance. The proposal that early macrophages are important for clearance implies that there is active phagocytosis of Six2^+^ cells. In the absence of this macrophage population there is a clear separation between a caudal Six2^+^ population and a more rostral population that is perhaps within the mesonephric field. Is it possible that macrophages generate signals that stimulate caudal migration of Six2^+^ cells rather than clearance? The presence of an increased total number of Six2^+^ cells in the DT experiments would help support the proposal that macrophages clear Six2^+^ cells via phagocytosis.In addition, it would be nice to know more about the aftereffects of failed clearance of Six2^+^ cells; for example, do ectopic nephrons form in the area corresponding to the mesonephros? Six2 cells secrete GDNF and other molecules, are there ectopic ureteric bud branches as a consequence of persistence of these cells?

Based on the localisation of macrophages within the rostral domain of Six2^+^ nephrogenic cells and the observation of Six2^+^ nuclei within the cell bodies of some macrophages at this rostral site we suggest our data are most consistent with the concept that macrophages actively phagocytose these cells (Figure 1G-J); however, we cannot reject the possibility that macrophages may also generate signals to stimulate caudal migration of Six2^+^ cells. Therefore, we have discussed these possibilities in a new paragraph in the Discussion section (paragraph three).

We also performed an additional staining / imaging experiment and prepared a new supplementary figure (Figure 1—figure supplement 3) showing high resolution images of macrophages engulfing rostral Six2^+^ nephron progenitor cells.

Regarding the after effects of failed clearance of Six2^+^ cells: we did not observe ectopic ureteric bud formation in any macrophage-depleted embryo (see new text in Results section paragraph five). In the absence of macrophages, ureteric bud development always occurred at an anatomically normal position; however, its development was delayed.

3) The proposal that a second wave of macrophages arrive via the vasculature needs to be better supported; for example, with movies, better quality images or alternately, the tone of the text needs to be softened.

We agree that we had not provided conclusive evidence that Gal3^+^ myeloid cells arrive as a later wave via the vasculature; we have softened the tone of our text to reflect this. We have, however, shown that a proportion of Gal3^high^ cells is carried within renal blood vessels and these data are present in Figure 4E-F’. We have now added additional images of intravascular Gal3^high^ cells in a new supplementary figure (Figure 4—figure supplement 2), to support this point even more strongly.

Other comments:1) The study makes a claim that a major role for the resident macrophage population is assisting in the joining of endothelial structures to form a mature and connected vasculature. This claim appears to rest on the quantification presented in Figure 7E-K. It is not clear how the feature measured in Figure 7K was reached. This refers to% of field covered by each CD31^+^ structure. How is a CD31^+^ structure defined here? In the non-depleted population this is a single plexus and it seems difficult to see how individual cells can be separated. The earlier data in Figure 7 showing phagocytosis is also very difficult to see as the critical panels are simply too small and the contrast, particularly on the blue channel, too high to see what is going on.

The% of the field of view covered by each CD31^+^ structure was determined by:

1) calculating the total% of the field covered by CD31^+^ structures

2) counting the number of isolated CD31^+^ structures per field

3) dividing the total% covered by CD31^+^ structures by the number of structures per field (see Materials and methods section)

The number of isolated CD31 structures per field was counted manually. The original count was non-blinded, so we re-performed this analysis using a blinded counter who was given coded samples to limit potential researcher bias (using the manual counting method as described in the Materials and methods section). The conclusions remained the same after the re-analysis, as the counts of the blinded counter were very similar to those of the original non-blinded researcher (data from this re-analysis are shown in Figure 7J-K).

The original representative field of view image shown for the ‘non-depleted’ / control-treated kidney had the fewest isolated structures of all samples, which may have made it difficult to interpret how the analysis was performed. We have changed this image to a more representative example (see new Figure 7F, anti-rat-IgG panels).

We have increased the size of the phagocytosis images in Figure 7A-B. In response to the comment, we have also reduced the contrast of the images in Figure 7A, particularly in the blue channel. Video 7 displays the individual z-planes of the relevant macrophages from Figure 7A-B, clearly showing that the phagocytosed cells are within the cell bodies of the macrophages of interest.

2) Lineage tracing experiments are required to definitively prove that Gal3^+^ macrophages seen at E14.5 are not just the F480^+^ macrophages that up-regulate Gal3 expression at this stage. Hence, this point should be toned down unless there is additional evidence supporting 2 lineages.

We agree that we have not excluded the possibility of interconversion occurring between F4/80^high^ and Gal3^high^ cells in the developing kidney. We have toned down our wording throughout the text when referring to the Gal3^high^ cells to avoid any suggestion that these cells have distinct developmental lineages from the F4/80^high^ cells.

3) The authors find that macrophages are localized between the Six2^+^ nephron progenitor cells and the nephric duct prior to the onset of ureteric bud outgrowth, and by E10.5 were localized in a rostral domain of Six2^+^ cells (this is the regressing mesonephros?) where the authors hypothesize they may be important for clearance.In terms of the broad community, it would be nice to link macrophage biology in the kidney with what is known in other systems. For example, what signals could be important for migration of early and late macrophage populations during development, are they conserved with pathways that function during pathogenesis or are known regulators likely to be involved? There might be some clues that can be gleaned from the single-cell RNA sequencing study.

We have added a new paragraph in the Discussion section (paragraph three) where we link our observation (that macrophages clear the rostral domain of the early nephron progenitor population) with previous studies that demonstrated the importance of cell clearance by macrophages in tissue development. Unfortunately, the single-cell RNA sequencing study was performed at too late an age (E18.5) to give us any indications about the signals (e.g. ligand-receptor pairs) that may be important for macrophage migration at the developmental stage when rostral nephron progenitors become diminished.